# $<\text{SO}\mathcal{G}_k>$: One LLM Token for Explicit Graph Structural Understanding

**Jingyao Wu**[1], **Bin Lu**[1*], **Zijun Di**[1], **Xiaoying Gan**[1], **Meng Jin**[1], **Luoyi Fu**[1], **Xinbing Wang**[1], **Chenghu Zhou**[2]

[1]Shanghai Jiao Tong University, [2]IGSNRR, Chinese Academy of Sciences

{wujingyao,robinlu1209,dzj75du,ganxiaoying,jinm,yiluofu,xwang8}
@sjtu.edu.cn, zhouch@lreis.ac.cn

## ABSTRACT

Large language models show great potential in unstructured data understanding, but still face significant challenges with graphs due to their structural hallucination. Existing approaches mainly either verbalize graphs into natural language, which leads to excessive token consumption and scattered attention, or transform graphs into trainable continuous embeddings (i.e., soft prompt), but exhibit severe misalignment with original text tokens. To solve this problem, we propose to incorporate one special token $<\text{SO}\mathcal{G}_k>$ to fully represent the **S**tructure **O**f **G**raph within a unified token space, facilitating explicit topology input and structural information sharing. Specifically, we propose a topology-aware structural tokenizer that maps each graph topology into a highly selective single token. Afterwards, we construct a set of hybrid structure Question-Answering corpora to align new structural tokens with existing text tokens. With this approach, $<\text{SO}\mathcal{G}_k>$ empowers LLMs to understand, generate, and reason in a concise and accurate manner. Extensive experiments on five graph-level benchmarks demonstrate the superiority of our method, achieving a performance improvement of 9.9–41.4% compared to the baselines while exhibiting interpretability and consistency. Furthermore, our method provides a flexible extension to node-level tasks, enabling both global and local structural understanding. The codebase is publicly available[1].

## 1 INTRODUCTION

In recent years, Large Language Models (LLMs) have demonstrated impressive capabilities in the general understanding of unstructured text (Guo et al., 2025). However, there are still significant limitations in how LLMs comprehend text-attributed graphs, such as knowledge graphs (Wang et al., 2025b; Lin et al., 2025), citation networks (Chen et al., 2024; Tang et al., 2024), transaction networks (Lei et al., 2025). Compared to LLM-as-Enhancer approaches (He et al., 2024a; Liu et al., 2024; Li et al., 2024b), LLM-as-Predictor methods (Ye et al., 2024; Tang et al., 2024; Chen et al., 2024; Huang et al., 2024) gain more widespread attention due to their generalizability across different level tasks and datasets/domains. Thus, for the serialized input format of LLMs, the core challenge in understanding graphs lies in how to feed the structure into the model in a complete, accurate, and efficient manner.

Existing works mainly transform the graph topology in two ways: one is Graph-to-Text, and the other is Graph-to-Embedding. (1) Graph-to-Text methods (Ye et al., 2024; Zhao et al., 2023b; Fatemi et al., 2024) directly verbalize graph topology with natural language and are widely used in prior work. Yet, due to the increasing scale of real-world graphs, depicting the complete topology in text consumes large amounts of tokens, which scatters LLM attention and impairs holistic structural understanding. In addition, existing work (Cao et al., 2025) reveals that LLMs exhibit strong structural hallucination, that even small permutations in node/edge orders can yield different output, inducing unstable structural understanding. (2) To reduce token overhead, Graph-to-Embedding methods (Tang et al., 2024; Chen et al., 2024; Huang et al., 2024; Liu et al., 2025a) compress graph

---

*Correspondence to: Bin Lu (robinlu1209@sjtu.edu.cn)

[1]The code of our project is available at https://github.com/Jingyao-Wu/SOG.

topology into continuous embeddings and project them into the LLM. However, it inevitably introduces a modality mismatch. The projected graph embeddings are poorly aligned with LLM inherent token space, leading to weak cross-modal compatibility and degraded structural comprehension. Therefore, it raises the following question: *How to concisely and accurately feed graph topology information into an LLM to avoid structural hallucination?*

To solve the above problem, we propose to incorporate **only one special token** for LLM, denoted as $<\text{SO}\mathcal{G}_k>$, to fully represent the $\underline{\text{S}}$tructure $\underline{\text{O}}$f $\underline{\text{G}}$raph within the same space with text tokens. $<\text{SO}\mathcal{G}_k>$ facilitates explicit topology input and structural information sharing. To be specific, we firstly propose a Topology-Aware Graph Structural Tokenizer to encode each graph topology faithfully and pull one structural token from discrete token vocabulary. This compact yet expressive token encapsulates critical topological information while demonstrating a highly selective property. To align $<\text{SO}\mathcal{G}_k>$ with LLMs textual input, we then construct a set of Hybrid Structure Question-Answering (QA) that combine text tokens and structural tokens within a task-agnostic context. After tuning on these novel corpora, it guides the structural tokens to live in the same space with text tokens, hereby promoting the LLM to seamlessly read and understand topology.

To summarize, the main contributions of our work are as follows:

- We propose to utilize only one structural token $<\text{SO}\mathcal{G}_k>$ obtained by a topology-aware graph tokenizer to effectively provide structure information to the LLM in a highly selective manner.
- To align the embedding space of two types of tokens, we design a set of hybrid structure QAs to instruct LLM to understand, generate and reason the structure, which harmonizes text and topology for a unified understanding.
- Extensive experiments on five graph-level benchmarks demonstrate that our method outperforms all the baselines with performance improvement ranging from 9.9% to 41.4%, while exhibiting strong interpretability and consistency. Additionally, our method can be effectively extended to node-level classification, showing $<\text{SO}\mathcal{G}_k>$ possesses both global and local structure perception.

## 2 RELATED WORK

Recently, LLMs show promising performance in understanding structured graph data, where the core issue is how to input non-Euclidean graph into serialized LLMs. Existing work can primarily be classified into two categories: *Graph-to-Text methods* and *Graph-to-Embedding methods*.

**Graph-to-Text methods** verbalize the graph using natural language, transforming the structure into a text sequence, which are widely adopted in existing works. The most straightforward methods (Ye et al., 2024; Wang et al., 2023) describe graph connectivity by enumerating edges and indirect in the form of "node A is connected to node B" along with edge attributes. Besides, Li et al. (2024a) and Wang et al. (2023) directly feed adjacency lists in text form into the LLM. To leverage the commonsense knowledge of LLMs, Talk Like a Graph (Fatemi et al., 2024) characterizes nodes and edges as various types of real-world connections, such as friendships, co-authorships, and character relationships. InstructGraph (Wang et al., 2024b) formulates a graph as a Python class with nodes and edges as attributes, leveraging LLMs' code reasoning abilities. To further improve traverse efficiency, GraphText (Zhao et al., 2023b) describes a $k$-hop neighbourhood structure of a central node with neighbouring features and labels, enlarging the receptive field of LLMs. LangTopo (Guan et al., 2024) innovatively constrains the LLM's output and aligns the LLM comprehension of text-formatted graph inputs with the GNN's structural understanding at quantization level. Recently, Dr.E (Liu et al., 2025b) summarize each hop information into multiple textual tokens in existing LLM vocabulary. However, all these Graph-to-Text methods require large token consumptions to fully depict the graph structure, scattering the model attention for an overall structural understanding. Moreover, LLMs are highly sensitive to variations in the order of node/edge description, leading to unstable structure reasoning (Cao et al., 2025).

**Graph-to-Embedding methods** compress the graph into embeddings and then project them into the token space of the LLM, which is also named 'soft prompt' in some works. GraphGPT (Tang et al., 2024) encodes textual and structural features separately, learns hybrid node embeddings via contrastive learning, and maps them into the LLM token space through an MLP-based projector. To reduce structural encoding complexity, LLaGA (Chen et al., 2024) adopts Laplacian embeddings and parameter-free node aggregation. Moving beyond node- or hop-level representations, G-retriever (He et al., 2024b) encodes retrieved subgraphs and pools them into graph-level embed-

Figure 1: The overall architecture for LLM understanding with structural token $<\text{SO}\mathcal{G}_k>$.

dings, which are then adapted to the LLM input space via an MLP. Several works further enhance structural expressiveness: TEA-GLM (Wang et al., 2024a) constructs contrastive views through edge deletion and node masking, while GraphLLM (Chai et al., 2023) employs a shared MLP to project graph embeddings into the input space of each LLM layer. Instead of using MLPs for projection, GraphAdapter (Huang et al., 2024) leverages a GNN-based projector to improve efficiency on text-attributed graphs. Beyond direct projection, TEA-GLM (Wang et al., 2024a) and GNP (Tian et al., 2024) additionally apply PCA-based mapping or attention-based pre-alignment before space alignment. In spite of these success in compressing structure information with less token consumptions than Graph-to-Text methods, Graph-to-Embedding methods still suffer from severe misalignment between LLM inherent token space and projected graph embeddings.

To sum up, our approach aims to utilize only one structural token, i.e., $<\text{SO}\mathcal{G}_k>$, sharing the same space with LLM inherent tokens, to fully characterize the graph structure. In this way, we follow a new **Graph-to-Token** paradigm, enabling a harmonious token mapping and understanding across graph structure and textual attributes within LLMs.

## 3 PRELIMINARY

Given a text-attributed graph, it contains graph structure $G = (V, E)$, corresponding text information $T$, and label $y$, where $V = \{v^1, v^2, \ldots, v^{|V|}\}$ is the set of nodes, and $E \subseteq V \times V$ is the set of edges. The adjacency matrix of $G$ is $A \in \{0, 1\}^{|V| \times |V|}$, with $A_{ij} = 1$ indicating the presence of edge $(v^i, v^j)$. We take an LLM $\mathcal{M}(\cdot; \mathcal{V}, \Theta)$ for graph understanding, where $\mathcal{V}$ denotes the vocabulary, and $\Theta$ denotes the parameters. In our work, we propose a group of structural tokens $\{<\text{SO}\mathcal{G}_1>, \cdots, <\text{SO}\mathcal{G}_K>\}$ to explicitly reflect graph overall structure information, where $K$ is the number of structural tokens. Each structural token $<\text{SO}\mathcal{G}_k>$ represents a prototype of pure graph structures. These structural token share a same token space with LLM original vocabulary $\mathcal{V}$, hereby forming a new vocabulary $\mathcal{V}' = \mathcal{V} \cup \{<\text{SO}\mathcal{G}_1>, \cdots, <\text{SO}\mathcal{G}_K>\}$.

Afterwards, to obtain the answer $\hat{y}$ using LLM as predictor, the input to $\mathcal{M}(\cdot; \mathcal{V}', \Theta)$ consists of: (1) task prompt $P$ with task requirements, desired output format and candidate labels, (2) textual graph attributes $T$, (3) corresponding structural token $<\text{SO}\mathcal{G}_k>$. Thus, our objective is let LLM $\mathcal{M}$ generates the output $\mathcal{O}$ containing the predicted label $\hat{y} \subset \mathcal{O}$ that matches the ground truth label $y$. Formally, this process can be denoted as follows:

$$\mathcal{O} = \mathcal{M}(\{P, T, <\text{SO}\mathcal{G}_k>\}|G; \mathcal{V}', \Theta). \quad (1)$$

This formulation allows structural information to be seamlessly integrated into the generative reasoning process of LLMs, thereby achieving more reliable inference on graph data.

## 4 METHODOLOGY

In this section, we formally introduce how to generate and utilize the structural token, i.e., $<\text{SO}\mathcal{G}_k>$, to enhance LLM's graph understanding in a two-stage manner as shown in Figure 1. Specifically, in the first stage, we propose a topology-aware graph structural tokenizer, which extracts the graph topology, encodes global information, and further projects into one structural token. In the second stage, multiple hybrid structure Question-Answer (QA) pairs are constructed and fed into the LLM to ensure the integration of text tokens and structural tokens in a unified space. Thereby, LLM can understand, generate and reason these tokens to obtain satisfied results across different graph tasks.

### 4.1 TOPOLOGY-AWARE GRAPH STRUCTURAL TOKENIZER

Given a text-attributed graph, the topology-aware graph structural tokenizer first extracts its graph topology $G(V, E)$, retaining only the pure structural information. To enhance the topology understanding, we assign each node $v^i \in V$ with a new structural attribute $t_s^i$ with a hierarchical

traversal strategy. An anchor node is first identified based on a node importance measure (e.g., node degree). Then, each node in the graph is located by its relative position to the anchor node, forming its node attribute such as 'first-hop neighbor #1', or 'second-hop neighbor #3'. The numbering of nodes within the same hop (e.g., #1, #3) is determined by ranking them according to the same node importance measure. In addition, we introduce a virtual global node to connect all nodes, serving as a pooling mechanism to obtain an overall representation. Finally, we adopt an off-the-shelf text encoder $f_T(\cdot)$ to embed the newly added attributes: $x_s^i = f_T(t_s^i)$, $X_s = [\,x_s^1; x_s^2; \ldots ; x_s^{|V|}; x_s^{\text{global}}\,] \in \mathbb{R}^{(|V|+1) \times d_s}$, where $d_s$ is the dimension of the text encoder. This hierarchical strategy provides a unique spatial coordinate and informative feature for each node, eliminating the vague description induced by node-invariant permutation.

Then, to aggregate node information across its multi-hop neighboring, we employ a graph neural network (GNN) $f_G(\cdot)$ as the encoding module (Kipf & Welling, 2017). Through message passing mechanism, GNN captures topology-aware information, mapping the node structural attributes $X_s$ to the latent representation $H_s$ in a continuous space, where $H_s = f_G(X_s)$, $H_s \in \mathbb{R}^{(|V|+1) \times d}$, and $d$ is the dimension of GNN encoder. The representation of the virtual global node $h_s^{\text{global}} \in \mathbb{R}^d$ is then taken as a graph-level structural representation. However, the continuous representation fails to align with discrete text tokens for LLMs, leading to unexpected structural hallucinations.

Therefore, we propose to map the continuous representations into $K$ discrete token vocabulary utilizing self-supervised topology reconstruction (Yang et al., 2024a), where $K$ is a hyperparameter that controls the size of the vocabulary. Specifically, we introduce a learnable vocabulary set denoted as $\mathcal{C} = \{c_1, c_2, \ldots, c_K\}$, which serves as a set of discrete latent embeddings for structural patterns. Each entry $c_j \in \mathcal{C}$ corresponds to a structural prototype learned during reconstruction task. For each node feature $h_s^i$, we compare and retrieve the nearest codebook entry $c_k = \mathbf{z}_e(h_s^i, \mathcal{C})$ based on Euclidean distance:

$$k = \arg \min_j \left\| h_s^i - c_j \right\|_2, \quad c_j \in \mathcal{C} = \{c_1, c_2, \ldots, c_K\}.$$

To achieve this, two loss functions, i.e., update loss $\ell_{\text{update}} = \left\| \text{sg}[h_s^i] - c_k \right\|_2^2$ and commitment loss $\ell_{\text{commit}} = \left\| h_s^i - \text{sg}[c_k] \right\|_2^2$, are utilized to find a bridge between $c_k$ and $h_s^i$. In particular, the vocabulary index corresponding to the global node is chosen as the structure vocabulary for the whole graph. To reflect its topology-awareness, the selected vocabulary entries enable the topology reconstruction with a lightweight decoder. Thus, we incorporate a shared decoder module $f_q(\cdot)$ to obtain reconstructed node features $\hat{X} = f_q(\mathbf{z}_e(H_s))$, where $\mathbf{z}_e(H_s) \subset \mathcal{C}^{|V|+1}$ denotes the selected vocabulary entries corresponding to the node features $H_s$. $\hat{X}$ is then used to reconstruct the adjacency matrix $\hat{A} = \hat{X}\hat{X}^T$, thereby computing the reconstruction loss between $\hat{A}$ and original $A$. Thus, the overall loss function consists of three components, and is further denoted as follows:

$$\mathcal{L} = \underbrace{\left\| A - \hat{A} \right\|_F^2}_{\text{Reconstruction loss}} + \underbrace{\left\| \text{sg}[H_s] - \mathbf{z}_e(H_s) \right\|_2^2}_{\text{Update loss}} + \beta \underbrace{\left\| H_s - \text{sg}[\mathbf{z}_e(H_s)] \right\|_2^2}_{\text{Commitment loss}},$$

where $\beta$ is the weighting hyperparameter.

### 4.2 Token Alignment with Hybrid Structure QAs

After obtaining the graph structural token, it requires to align with original LLM text tokens for a unified understanding. To solve this problem, we construct hybrid structure QAs to assign reasonable embeddings for newly-added structural tokens and tune the LLM encoder for context awareness. We design 3 types of corpora for the structure QAs, which are illustrated as follows:

1. $k$-**Nearest Token Neighbour Matching**: The question is *"Here is the target structural token $<SO\mathcal{G}_i>$, and its five nearest graph structural tokens are:"*. Afterwards, the ideal answer is its Top-$k$ nearest tokens based on vector cosine similarity within the vocabulary. This QA corpus ensures a local agreement with that the closer the discrete structural representation is, the closer the LLM token embedding is.

2. **True/False Structure Similarity Judgment**: The question is *"Here are two tokens $<SO\mathcal{G}_i>$ and $<SO\mathcal{G}_j>$, judge whether they represent similar structures or not."* If the cosine similarity

between two corresponding global node embeddings in the continuous space is greater than a predefined positive threshold, the answer is "similar"; if it is smaller than a negative threshold, the answer is "dissimilar". This QA corpus leverages the idea of contrastive learning to clarify the boundary of structural similarity in token embedding space.

3. **Description-Token Pairs Matching**: The question is *"Here is the target graph: node A and node B is connected; node A and node C is connected ...... The corresponding graph structural token is:"*. The answer is its true structural token. This QA corpus directly link the text token understanding with the structural token understanding, encouraging LLM to explicitly unify two tokens in the same "brain".

Through these QAs, we conduct supervised fine-tuning on the LLM. (1) To maintain pretrained knowledge, we freeze the pre-existing text tokens, and only update the token embeddings for newly-added structural tokens. (2) To further improve the topology-awareness, we employ a lightweight parameter adaptation technique, i.e., LoRA (Hu et al., 2022), to efficiently inject structural information into the model. Notably, in this fine-tuning process, the constructed QAs solely depend on the graph structure, neglecting the original textual information or downstream tasks. Therefore, it offers strong generalizability and scalability across different graph tasks and domain information.

## 4.3 DOWNSTREAM APPLICATION WITH $<$SO$\mathcal{G}_k>$

To guide LLM for different downstream applications, we conduct a task-specific fine-tuning stage, hereby transforming the task into a text generation paradigm. Specifically, we first input a system prompt $S$ as a general guidance: "Classify target graph into the correct category based on its graph topology (i.e. structural token) and textual attributes. Each structural token represents a unique graph pattern (e.g., a prototype of similar molecular structures)." Following $S$, we append a user prompt to concretize the input information and application scenario. The user prompt of LLM consists of the following three parts: (1) Task Prompt $P$: Explicitly guide the LLM to understand task objective, such as "[Task] Predict whether the molecule is toxic." or "[Task] Classify the central paper node by assigning it the correct label from the set of candidates..." (2) Textual Graph Attributes $T$ : Provide original semantic features and background information about the graph, such as '[Molecule] The molecule's SMILES expression is...' or '[Paper] The center node represents a paper titled...' (3) Assigned Structural Token $<$SO$\mathcal{G}_k>$: Represent the graph topology using "[Structural Token] $<$SO$\mathcal{G}_k>$". During task-specific fine-tuning, the ground-truth label $y$ for the particular downstream application is incorporated after the LLM input, i.e.system prompt and user prompt, as supervised signal to guide response generation. The loss function is denoted as follows:

$$\mathcal{L} = -\sum_t \log p_\Theta(y_t \mid y_{<t}, P, T, <\text{SO}\mathcal{G}_k>),$$

where $t$ indexes the position in the output sequence. In this way, downstream applications can further align text/structure understanding with different domain tasks.

## 5 EXPERIMENT

In this section, we empirically evaluate the effectiveness and structural contributions of $<$SO$\mathcal{G}_k>$ via extensive experiments by addressing the following research questions (**RQ**s):

- **RQ1**: How does $<$SO$\mathcal{G}_k>$ perform compared to other baselines in graph classification tasks?
- **RQ2**: How effective are the proposed structural tokens $<$SO$\mathcal{G}_k>$?
- **RQ3**: How interpretable are the structural tokens? Do they maintain alignment with text tokens?
- **RQ4**: Can $<$SO$\mathcal{G}_k>$ extend to improve LLM structural understanding on node-level tasks?
- **RQ5**: How sensitive is $<$SO$\mathcal{G}_k>$ to key structural tokenizer design choices?

### 5.1 EXPERIMENT SETUP

**Datasets**. To evaluate performance, we perform graph-level classification tasks on 5 datasets from MoleculeNet (Wu et al., 2017), including BBBP, Tox21, ClinTox, HIV, and BACE, covering toxicology, clinical research, and pharmaceutical chemistry domains and encompassing 17 subtasks. The detailed information is summarized in Appendix A.1.

Table 1: Performance comparison across five datasets, where **bold** indicates the best performance and underlined indicates the second-best.

| LLM Backbone / Method | | BBBP (↑) | Tox21 (↑) | ClinTox (↑) | HIV (↑) | BACE (↑) |
|---|---|---|---|---|---|---|
| GPT-4 | Zero-shot | 61.5 | 55.2 | 51.6 | 65.9 | 62.5 |
| GPT-4o | Zero-shot | 57.0 | 36.0 | 51.8 | 54.7 | 38.5 |
| Deepseek-R1 | Zero-shot | 63.6 | 60.1 | 48.2 | 50.5 | 52.7 |
| LLaMA3-70B | Zero-shot | 60.1 | 44.9 | 48.8 | 58.3 | 50.9 |
| Galactica-120B | Zero-shot | 66.1 | 68.9 | 82.6 | 74.5 | 61.7 |
| Non-LLM Molecule Approach | GPF-AttrMasking | 58.8 | 66.0 | - | 71.3 | 62.2 |
| | GPF-GCL | 56.5 | 63.6 | - | 49.3 | 52.1 |
| | GIMLET | 59.4 | 61.2 | - | 66.2 | 69.6 |
| | MolCA-S | 62.5 | 66.6 | - | 69.0 | 62.8 |
| | MolCA-GS | 63.6 | 68.5 | - | 72.7 | 63.9 |
| LLaMA3-3B | Zero-shot | $50.2 \pm 3.4$ | $48.5 \pm 2.7$ | $50.2 \pm 2.5$ | $51.8 \pm 0.8$ | $52.7 \pm 1.8$ |
| | Few-shot (Random) | $46.5 \pm 3.8$ | $49.9 \pm 10.2$ | $57.8 \pm 6.2$ | $49.7 \pm 16.7$ | $68.2 \pm 17.5$ |
| | Few-shot (Morgan Sim) | $49.7 \pm 2.5$ | $51.0 \pm 8.7$ | $53.3 \pm 10.0$ | $42.0 \pm 14.7$ | $49.7 \pm 0.6$ |
| | LoRA SFT | $60.2 \pm 1.2$ | $50.8 \pm 0.8$ | $63.8 \pm 2.8$ | $51.5 \pm 0.7$ | $50.5 \pm 2.9$ |
| | Soft Prompt | $27.9 \pm 1.7$ | $39.9 \pm 1.9$ | $35.0 \pm 2.0$ | $33.6 \pm 0.5$ | $28.5 \pm 0.9$ |
| | **<SO$\mathcal{G}_k$> (Ours)** | $\mathbf{76.9 \pm 3.1}$ | $\mathbf{83.4 \pm 3.3}$ | $\underline{85.5 \pm 3.7}$ | $\underline{75.7 \pm 1.6}$ | $63.3 \pm 4.2$ |
| LLaMA2-7B | Zero-shot | $50.0 \pm 0.0$ | $48.4 \pm 3.3$ | $44.0 \pm 2.6$ | $50.0 \pm 0.0$ | $49.9 \pm 0.4$ |
| | Few-shot (Random) | $49.4 \pm 0.6$ | $49.3 \pm 2.2$ | $50.3 \pm 1.7$ | $46.9 \pm 0.8$ | $47.8 \pm 1.8$ |
| | Few-shot (Morgan Sim) | $51.3 \pm 0.8$ | $51.1 \pm 2.5$ | $56.8 \pm 6.3$ | $50.4 \pm 0.7$ | $53.2 \pm 1.5$ |
| | LoRA SFT | $55.0 \pm 0.8$ | $60.6 \pm 3.4$ | $59.3 \pm 3.7$ | $78.1 \pm 2.2$ | $61.7 \pm 2.7$ |
| | Soft Prompt | $53.7 \pm 0.8$ | $49.8 \pm 1.5$ | $49.7 \pm 4.7$ | $44.9 \pm 0.2$ | $49.4 \pm 2.3$ |
| | **<SO$\mathcal{G}_k$> (Ours)** | $\underline{66.4 \pm 2.7}$ | $\underline{72.4 \pm 3.1}$ | $\mathbf{94.3 \pm 0.1}$ | $\mathbf{83.2 \pm 1.9}$ | $\mathbf{98.4 \pm 0.8}$ |

**Evaluation Metrics**. We adopt generative question answering method to obtain the result from LLM, and evaluate by text matching to determine whether the correct answer appears in the response. The maximum output token length is limited to 100. The LLM sampling parameters are set with temperature of 0.6 which strikes a balance between randomness and coherence, while limiting the token selection to those with a cumulative probability mass of 90%. To quantitatively evaluate the classification results, we use AUC-ROC to assess the performance of binary graph classification tasks, as most tasks exhibit a significant class imbalance. Each experiment is run three times. and we all report the average performance. The detailed information is summarized in Appendix A.2.

**Baselines**. We compare our proposed method against 20 baselines, which are generally divided into three categories: (1) LLMs with extensive parameters ($\geq$ 70B), including GPT-4 (OpenAI, 2023), GPT-4o (Hurst et al., 2024), Deepseek-R1 (Guo et al., 2025), LLaMA3-70B (Dubey et al., 2024), Galactica-120B (Taylor et al., 2022); (2) non-LLM approaches, including GPF-AttrMasking (Fang et al., 2023; Hu et al., 2020b), GPF-GCL (Fang et al., 2023; You et al., 2020), GIMLET (Zhao et al., 2023a), MolCA-S (Liu et al., 2023b), MolCA-GS (Liu et al., 2023b); (3) Methods on LLM with few parameters (i.e. LLaMA2-7B (Touvron et al., 2023), LLaMA3-3B (Dubey et al., 2024)), including zero-shot, few-shot (Brown et al., 2020), LoRA-based supervised fine-tuning (LoRA SFT) (Hu et al., 2022), and soft-prompt (Perozzi et al., 2024). Detailed information refer to Appendix A.3.

**Implementation Details.** We illustrate the implementation details of our method in Appendix B.1. Meanwhile, the prompt designs for individual downstream tasks are presented in Appendix B.4, and additional training configurations and details are documented in Appendix B.2.

## 5.2 OVERALL PERFORMANCE (RQ1)

The experiment results on five graph-level datasets are shown in Table 1. We have the following observations: (1) <SO$\mathcal{G}_k$> consistently outperforms all the baselines, demonstrating a satisfied performance improvement from 9.9% to 41.4%. In the BBBP and Tox21 datasets, our method rank first and second in performance on the 3B and 7B models, respectively. On the remaining three datasets, our method on the 7B model achieves the best performance. Notably, on BACE dataset, <SO$\mathcal{G}_k$> attains an AUC-ROC of 98.4, with a significant performance gain over sub-optimal model. Due to the varying levels of comprehension difficulty and the number of training samples across different datasets, the increase in model parameter scale does not result in direct performance improvements on individual datasets. (2) Compared to non-LLM methods (GPF, GIMLET, MolCA),

Table 2: Ablation of different structural token <SO$\mathcal{G}_k$> on five datasets.

| Model | Ablation Study | BBBP | Tox21 | ClinTox | HIV | BACE |
|---|---|---|---|---|---|---|
| LLaMA3-3B | (M1a) w/o Structural Token | 61.2 | 72.3 | 81.8 | 53.0 | 53.2 |
| | (M1b) Static Structural Token | 60.7 | 67.8 | 52.5 | 54.6 | 54.3 |
| | (M1c) Random Structural Token | 57.3 | 62.2 | 52.8 | 55.3 | 55.8 |
| | Ours | **76.9** | **83.4** | **85.5** | **75.7** | **63.3** |
| LLaMA2-7B | (M1a) w/o Structural Token | 61.3 | 66.4 | 79.5 | 69.6 | 94.2 |
| | (M1b) Static Structural Token | 64.7 | 69.1 | 92.0 | 81.5 | 94.2 |
| | (M1c) Random Structural Token | 62.1 | 68.6 | 88.5 | **83.4** | 96.7 |
| | Ours | **66.4** | **72.4** | **94.3** | 83.2 | **98.4** |

large parameter LLMs, such as Galactica-120B, exhibit strong text comprehension and achieve better results on some datasets even in a zero-shot setting. However, due to limitations in structural understanding, their performance improvements are constrained. Our proposed structural token approach requires only a single token to surpass the performance of large models on smaller parameter LLMs (3B and 7B), highlighting our effectiveness. Meanwhile, our approach is inherently more cost-effective and resource-efficient. (3) For the baseline methods of the 3B/7B large language models, few-shot approach exhibit a large variance. For instance, the 3B model shows standard deviations as large as 16.7 on HIV and 17.5 on BACE, making the results unstable and unreliable. LoRA SFT method improves performance by explicitly fine-tuning parameters to adapt to the corresponding datasets. However, the limitations of the Graph-to-Text input method for LLMs constrain the model's improvement. Soft prompt suffers from severe misalignment issues, and in some cases even introduces negative effects for smaller models like 3B, with performance drops ranging from 8.6 to 24.2 percentage point over zero-shot performance. By comparison, our approach shows consistent performance improvement across different model scales, bridging the gap between LLMs textual reasoning and topology understanding using one single token effectively and efficiently.

## 5.3 ABLATION STUDY (RQ2)

**How do structural tokens impact the effectiveness of the model?** First of all, Figure 2 shows a toy example on the performance variation across the selection of different structural token <SO$\mathcal{G}_k$> for one specific graph instance. The label of this graph is "True," and we plot the generation probability of the LLM outputting 'True' when different structural tokens are input. When our model selects the expected <SO$\mathcal{G}_{157}$>, the probability of outputting the correct answer is maximized. In contrast, other structural token choices lead to highly divergent results, indicating that the structural tokens we generate are highly selective, and selecting the appropriate one can significantly enhance the model's performance. Furthermore, we conduct the ablation study with three different variants as shown in Table 2: (1) **w/o Structural Token**: Removing the structural token <SO$\mathcal{G}_k$> from LLM input and keep only task prompt and textual graph attributes. (2) **Random Structural Token**: Replace the specific <SO$\mathcal{G}_k$> with a random one. (3) **Static Structural Token**: Input a static structural token without discrimination. We observe that any variants all lead to a performance drop, confirming that structural tokens play a crucial role in guiding LLMs to understand graph topology. The statistical indicators demonstrate the highly selective nature of structural tokens: only the correct token generated by our topology-aware graph structural tokenizer is able to faithfully convey the correct topology information of the corresponding graph and thereby yield consistent improvements in downstream tasks, whereas random or static substitutions fail to provide meaningful guidance. This observation provides strong evidence that our LLM has learned to associate the specific topology-aware token with the corresponding graph structures. Consequently, only the properly mapped token can reliably transmit graph topology and drive accurate predictions.

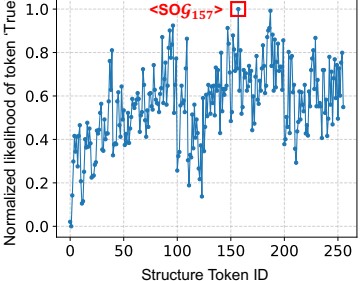

Figure 2: A toy example of performance variation across the selection of different structural tokens. More examples are shown in Figure 7, 8, 9 in Appendix C.

**How does the use of different hybrid QA methods for instruction tuning affect the performance of the model?** Figure 3 shows the model performance of instruction fine-tuning using three

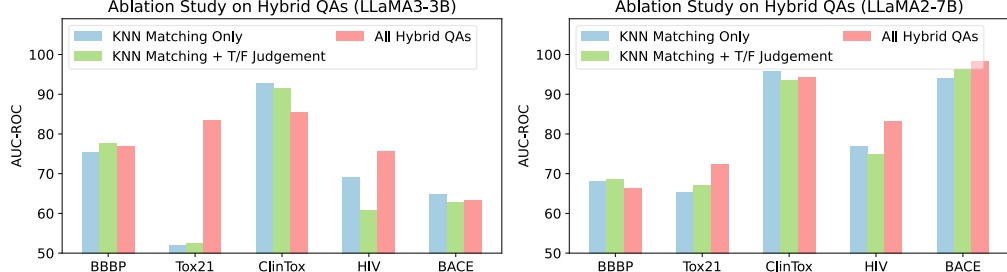

Figure 3: Ablation of tuning the LLM with different hybrid QAs corpora on five datasets.

different QA corpora. We find that the model generally performs better when using all hybrid QAs, particularly with Tox21 achieving remarkable results on the 3B model. At the same time, we observe that fine-tuning with description-token matching occasionally leads to a slight decrease in performance. This is because it requires the LLM to perform fine-grained structural characterization, which increases the learning difficulty of the model in certain scenarios.

## 5.4 CASE STUDY & MODEL ANALYSIS (RQ3)

**Does the explicit structural token have interpretability?** To verify this, we select the first 50 structural tokens, and as shown in Figure 4, we present the heatmap of their correlation matrix. We observe that the diagonal entries are 1.0, and beyond trivial self-similarity, the matrix is dominated by relatively weak (around 0.6-0.7) off-diagonal entries; furthermore, some token (e.g. token $<\text{SO}\mathcal{G}_{11}>$) exhibits uniformly low correlation with all others. This pattern indicates low redundancy among $<\text{SO}\mathcal{G}_k>$s and suggests a compact, near-orthogonal tokenization of graph topology.

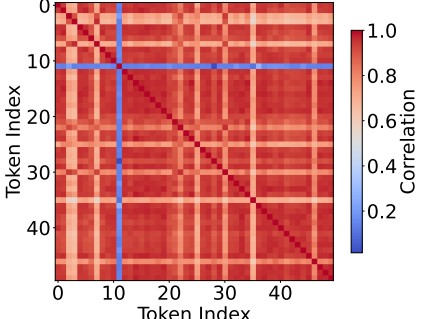

Figure 4: Correlation heatmap of first 50 structural tokens.

**Does the selection of structure tokens share similar structural information across datasets?** We examine the selection of structure tokens and their corresponding structures in different molecule datasets, as shown in Figure 5. Each row represents a set of examples in which the corresponding structure token is selected from different datasets. We find that these molecular structures share the same Bemis–Murcko scaffold (first column), which was extracted using the RDKit open-source toolkit (https://www.rdkit.org/). It reveals that molecules sharing the same scaffold are consistently mapped to the same structural token. This scaffold-level consistency confirms that $<\text{SO}\mathcal{G}_k>$ can capture core topological motifs, thereby providing a concise and non-redundant compression of graph structure.

**Do structure tokens and text tokens share the same embedding space?** Here we compare the relationship of structural tokens, text tokens, and graph soft prompts of LLaMA2-7B. We randomly sample 300 graphs and generate their soft prompts via the GCN, then extract embeddings for both structural tokens and frequent text tokens from the LLM's vocabulary. We project all three ones into 2D using PCA followed by t-SNE. Figure 6 visualizes that soft prompts and text tokens occupy clearly separable regions, revealing severe misalignment between graph-derived and language-derived embeddings. In contrast, structural tokens are embedded within the neighborhood of text tokens, effectively bridging the graph and language spaces. This pattern suggests that structural tokens uniformly inject the topology knowledge in the LLM's token space, thereby enabling LLM to read and reason over graph structural information.

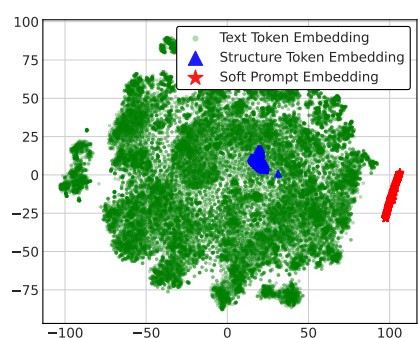

Figure 6: t-SNE visualization of structural tokens, text tokens, and graph-derived soft prompt embeddings

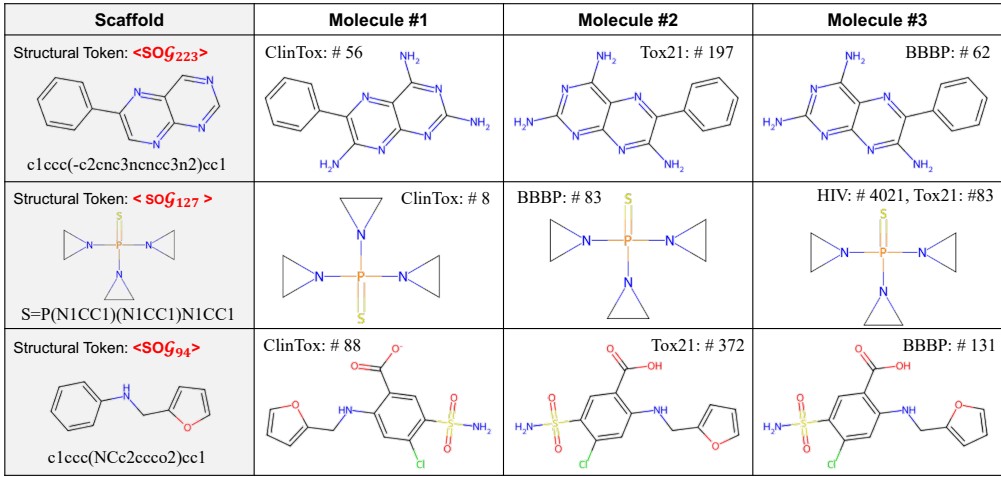

Figure 5: Molecules sharing the same scaffold consistently select the same structural token. More examples are shown in Figure 10, Figure 11 in Appendix C.

## 5.5 EXTENSION TO NODE-LEVEL CLASSIFICATION (RQ4)

In addition to the graph-level task, our method is easy to extend to node-level classification. Besides the global virtual node, each node in the graph is also assigned a corresponding structural token. Thus, we inject the node-level structural token to the LLM, and compare its performance on two widely-adopted benchmarks, Cora and Pubmed (Hu et al., 2020a; Li et al., 2024c). The detailed operation please refer to Appendix B.3. A series of baselines are compared for comprehensive comparison , including non-LLM approaches (DGI (Velickovic et al., 2019), GraphMAE (Hou et al., 2022)), general LLMs with extensive parameters (LLaMA3-70B (Dubey et al., 2024), GPT-3.5-turbo (Ouyang et al., 2022), GPT-4o (Hurst et al., 2024), DeepSeek-chat (Guo

Table 3: Performance comparison on node classification, where **bold** indicates the best performance and underlined indicates the second-best.

| Model | Cora | | Pubmed | |
|---|---|---|---|---|
| | Acc (↑) | F1 (↑) | Acc (↑) | F1 (↑) |
| DGI | 17.50 | 12.44 | 44.88 | 38.72 |
| GraphMAE | 27.08 | 23.66 | 22.03 | 15.65 |
| LLaMA3 (70B) | 67.99 | 68.05 | 77.00 | 64.18 |
| GPT-3.5-turbo | 65.67 | 63.22 | 75.99 | 69.90 |
| GPT-4o | 68.62 | 68.49 | 77.96 | 71.79 |
| DeepSeek-chat | 65.62 | 65.77 | 79.23 | 74.30 |
| Emb w/ NA | 63.59 | 58.23 | 74.66 | 73.15 |
| OFA | 23.11 | 23.30 | 46.60 | 35.04 |
| ZEROG | 62.52 | 57.53 | 79.08 | 77.94 |
| GraphGPT | 24.90 | 7.98 | 39.85 | 20.07 |
| Ours (LLaMA3-3B) | **91.58** | **78.62** | **97.46** | 85.50 |
| Ours (LLaMA2-7B) | 88.80 | 71.28 | 96.27 | **89.64** |

et al., 2025)) and graph specific LLM methods (Emb w/ NA (Li et al., 2024c), OFA (Liu et al., 2024), ZEROG (Li et al., 2024b), GraphGPT (Tang et al., 2024)).

As shown in Table 3, our method consistently achieves state-of-the-art performance across all datasets. Remarkably, the proposed $<SO\mathcal{G}_k>$ with only 3B parameters surpasses much larger models such as GPT-4o and DeepSeek-chat, yielding gains of 22.96 percentage point over GPT-4o and 25.96 percentage point over DeepSeek-chat in accuracy. Compared with ZEROG, which shows good performance among GraphLLM baselines, our approach achieves a 23% gain in accuracy and a 15% improvement in F1. The node-level experiment demonstrates that structural tokens exhibit both global and local structure perspective, due to our task-agnostic alignment strategy.

## 5.6 HYPERPARAMETER & DESIGN SENSITIVITY ANALYSIS (RQ5)

We conduct structural hyperparameter studies to examine how key design choices affect the effectiveness of $<SO\mathcal{G}_k>$. First, we evaluate different vocabulary sizes $K$ (e.g., 64, 128, 256, 512) and results are shown in Table 4. We observe that $K=256$ achieves the best overall performance. Smaller values fail to adequately distinguish diverse structural patterns, while excessively large values tend to overfit and reduce generalization; thus, $K=256$ provides a favourable balance between expres-

Table 5: Comparison of Anchor Node Selection Strategies.

| Task | random | pagerank | betweeness | degree* |
|------|--------|----------|------------|---------|
| BBBP_p_np | 74.4 | **82.2** | 80.9 | 76.9 |
| HIV_HIV_active | 72.0 | 75.4 | 62.3 | **79.7** |
| BACE_Class | 63.5 | 64.3 | 65.5 | **98.4** |
| ClinTox_FDA_APPROVED | 92.4 | **99.6** | 98.0 | 77.8 |
| Tox21_NR-AR | 69.6 | 59.6 | 69.7 | **76.4** |
| Tox21_NR-AR-LBD | 66.6 | 58.7 | 64.0 | **76.0** |
| Tox21_SR-p53 | **66.6** | 54.6 | 56.4 | 63.3 |
| avg. | 72.2 | 70.6 | 71.0 | 78.4 |

siveness and robustness. Our choice of $K$=256 provides the most appropriate capacity to model the structural diversity exhibited in our benchmarks.

In addition, we conduct sensitivity analysis on anchor node selection. We adopt node degree (*) by default, and compare it with alternatives, i.e., PageRank (Yang et al., 2024b), Betweenness Centrality (Xiang et al., 2023) and random selection, to assess their impact on structural encoding and downstream performance, as shown in Table 5. The degree-based strategy remains the most stable and effective in most benchmarks. PageRank offers slight improvements on BBBP and ClinTox due to its stronger capture of global centrality in sparse molecular graphs, while random selection often shifts the structural center unpredictably, resulting in unstable encoding and degraded performance.

Table 4: Effect of Structural Vocabulary Size $K$ on Model Performance.

| Dataset | Structural Vocab Size | | | |
|---------|------|------|------|------|
| | 64 | 128 | 256* | 512 |
| BBBP | 84.6 | 71.8 | 76.9 | 96.2 |
| ClinTox | 91.8 | 90.1 | 76.9 | 90 |
| HIV | 52.7 | 51.9 | 79.7 | 55.9 |
| BACE | 69.8 | 62.3 | 98.4 | 85.7 |
| Tox21 | 52.6 | 53.4 | 81.6 | 59.8 |
| **avg.** | **60.1** | **59.3** | **81.7** | **66.8** |

## 6 CONCLUSION

In this paper, we propose to utilize one structural token $<\text{SO}\mathcal{G}_k>$ for LLM explicit topology comprehension both concisely and accurately. Our method addresses critical issues of token inefficiency and cross-modal misalignment of the two mainstream paradigms in LLM structural understanding. We empirically validate the effectiveness of $<\text{SO}\mathcal{G}_k>$ on both graph-level and node-level benchmarks, demonstrating that our structural tokens are not only well aligned with the LLM's native token space, but also highly selective in capturing graph distinct structural patterns. In the future, we will focus on elaborating its capabilities on more diverse graph data, such as dynamic graphs and heterogeneous graphs.

ACKNOWLEDGMENTS

The authors of this paper were supported by National Natural Science Foundation of China (No. T2421002, 92579211, 62272301, 623B2071, 62525209), Postdoctoral Fellowship Program of CPSF under Grant Number No. GZB20250806 and the AI for Science Seed Program of Shanghai Jiao Tong University (project number 2025AI4S-QY01).

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

# A  ADDITIONAL DETAILS ON EXPERIMENT SETUP

## A.1  DATASETS

MoleculeNet is a benchmark suite designed to standardize datasets and create a consistent evaluation framework for machine learning models in computational chemistry (Wu et al., 2017). The following five datasets (BBBP, BACE, HIV, CinTox, Tox21) all come from this benchmark. We follow the dataset selection and splitting criteria specified in GIMLET (Zhao et al., 2023a). As for downstream tasks, we divide the datasets into a ratio of 8:1:1 and report the results on the test sets. There are single-task and multi-task setups, and multiple tasks from the same dataset differ only in task labels while sharing identical molecules. Data statistics is shown in Table 6.

**BBBP**  The BBBP (Blood-Brain Barrier Permeability) dataset is part of the MoleculeNet collection. In particular, it is used to predict whether a given chemical compound can permeate the blood-brain barrier (BBB). BBBP contains binary labels for compounds that indicate whether they can cross the blood-brain barrier or not, 1 for compounds that can pass through the BBB and 0 for those that cannot. These compounds are represented by SMILES expressions. The dataset includes 2,039 compounds with various chemical features, including molecular properties such as atom type, bond type, molecular weight, and so on. However, they are not used as assistant information in the model input in this task. The dataset is widely used for drug development research. This is important in the design of drugs aimed at treating brain-related diseases, as only compounds that can cross the BBB are able to have an effect on the central nervous system.

Table 6: Graph-level classification datasets statistics.

| dataset | # total samples | | | positive prop.(%) | | | negative prop.(%) | | |
|---|---|---|---|---|---|---|---|---|---|
| | train | valid | test | train | valid | test | train | valid | test |
| BACE_Class | 1210 | 151 | 152 | 42.6 | 55.6 | 60.5 | 57.4 | 44.4 | 39.5 |
| BBBP_p_np | 1631 | 204 | 204 | 82.2 | 54.9 | 52.5 | 17.8 | 45.1 | 47.6 |
| ClinTox_CT_TOX | 1184 | 148 | 148 | 8.0 | 4.7 | 6.8 | 92.0 | 95.3 | 93.2 |
| ClinTox_FDA_APPROVED | 1184 | 148 | 148 | 93.3 | 96.0 | 93.9 | 6.7 | 4.1 | 6.1 |
| HIV_HIV_active | 32896 | 4112 | 4112 | 3.8 | 2.0 | 3.2 | 96.3 | 98.0 | 96.8 |
| Tox21_NR-AR | 6258 | 782 | 783 | 4.0 | 4.0 | 3.5 | 96.0 | 96.0 | 96.6 |
| Tox21_NR-AR-LBD | 6258 | 782 | 783 | 3.1 | 3.2 | 2.4 | 96.9 | 96.8 | 97.6 |
| Tox21_NR-AhR | 6258 | 782 | 783 | 9.4 | 11.1 | 11.8 | 90.6 | 88.9 | 88.3 |
| Tox21_NR-Aromatase | 6258 | 782 | 783 | 3.3 | 5.8 | 6.0 | 96.7 | 94.3 | 94.0 |
| Tox21_NR-ER | 6258 | 782 | 783 | 10.3 | 9.6 | 8.9 | 89.7 | 90.4 | 91.1 |
| Tox21_NR-ER-LBD | 6258 | 782 | 783 | 4.8 | 3.7 | 2.7 | 95.2 | 96.3 | 97.3 |
| Tox21_NR-PPAR-gamma | 6258 | 782 | 783 | 2.1 | 4.1 | 2.8 | 97.9 | 95.9 | 97.2 |
| Tox21_SR-ARE | 6258 | 782 | 783 | 11.5 | 13.6 | 15.1 | 88.5 | 86.5 | 84.9 |
| Tox21_SR-ATAD5 | 6258 | 782 | 783 | 3.1 | 4.5 | 4.2 | 96.9 | 95.5 | 95.8 |
| Tox21_SR-HSE | 6258 | 782 | 783 | 4.5 | 5.6 | 6.0 | 95.5 | 94.4 | 94.0 |
| Tox21_SR-MMP | 6258 | 782 | 783 | 11.4 | 14.2 | 12.3 | 88.6 | 85.8 | 87.7 |
| Tox21_SR-p53 | 6258 | 782 | 783 | 4.4 | 9.6 | 9.2 | 95.6 | 90.4 | 90.8 |

**BACE** The BACE dataset from MoleculeNet is designed for binary classification, aiming to predict whether a molecule will bind to the human $\beta$-secretase, which is an aspartic acid protease important in the formation of myelin sheaths in peripheral nerve cells. In the BACE binary classification task, a label of 1 indicates that the molecule is a BACE inhibitor, while a label of 0 signifies that the molecule is not a BACE inhibitor. The dataset contains 1513 molecules, each with molecular features, i.e. their SMILES expressions. It is widely used for evaluating machine learning models in drug discovery.

**HIV** The HIV dataset in MoleculeNet aims to predict whether a drug can inhibit HIV replication. This dataset consists of 41120 molecules with features , e.g. smile expressions and their corresponding activity against the HIV virus. The task is binary classification, a label of 1 indicates that a given drug molecule can inhibit HIV replication, while a label of 0 signifies that it can not. The dataset is widely used for evaluating machine learning models in the context of drug discovery and molecular bioinformatics, particularly in the realm of HIV research and drug development.

**ClinTox** The ClinTox dataset is derived from drug discovery studies and collected in MoleculeNet. It focuses on clinical trial outcomes and drug safety evaluation, containing 1480 compounds annotated with two binary classification tasks. One task, FDA-APPROVED, predicts whether a drug molecule is approved by the U.S. Food and Drug Administration (FDA), where label 1 denotes FDA-approved drugs and 0 denotes non-approved drugs. The other task, CT-TOX, determines whether a drug molecule has failed clinical trials due to toxicity, with 1 for failure and 0 for success and continuation. This dataset is widely used for toxicity prediction and drug safety assessment.

**Tox21** The Tox21 dataset originates from the Toxicology in the 21st Century (Tox21) program and is included in MoleculeNet. It is designed for evaluating the human toxicity of chemical compounds across 12 biological targets. It consists of 7,823 molecules, each annotated with 12 binary labels indicating whether the compound activates or inhibits a specific target. Specifically, the nuclear receptor (NR) tasks include NR-AR, NR-AR-LBD, NR-AhR, NR-Aromatase, NR-ER, NR-ER-LBD, and NR-PPAR-gamma, each requiring the prediction of whether a molecule activates or inhibits the corresponding receptor. The stress response (SR) tasks include SR-ARE, SR-ATAD5, SR-HSE, SR-MMP, and SR-p53, which assess whether a compound induces responses such as antioxidant signaling, DNA damage, heat shock, mitochondrial dysfunction, or p53-mediated stress pathways. All tasks are formulated as binary classification problems, where 1 denotes activation and 0 denotes no activation. This dataset has become a widely adopted benchmark in computational toxicology and drug safety evaluation.

A.2 EVALUATION METRICS

To quantitatively evaluate the graph classification results, we use AUC-ROC (Area Under the Receiver Operating Characteristic Curve) to assess the performance of binary graph classification tasks, as most tasks exhibit a significant class imbalance. AUC-ROC measures the model's ability to distinguish between classes by plotting the true positive rate (TPR) against the false positive rate (FPR) across different thresholds. The higher the value, the better the model's discrimination ability.

For the calculation of the AUC-ROC metric in practice, the maximum output token length is limited to 100. To generate output, the LLM sampling parameters are set with temperature of 0.6 which strikes a balance between randomness and coherence, while limiting the token selection to those with a cumulative probability mass of 90%. Then we parse the generated text to extract answers. Specifically, we perform matching by defining a positive set (eg.["yes","true", "active", "approved"]) and a negative set (eg.["no", "false", "inactive", "rejected", "not approved"]). If any word/phrase from the positive set appears in the generated text, the prediction is assigned 1; otherwise, if any one from the negative set appears, it is assigned 0. To mitigate substring conflicts (e.g., positive phrase embedded in negations), we enforce matching precedence: negatives are evaluated first, and positives are considered only in the absence of a negative match. The resulting predictions are paired with ground-truth labels (0/1) to compute AUC-ROC.

For multi-task evaluation, the reported mean is obtained by averaging the mean performance across individual tasks. Assuming independence between tasks, the overall standard deviation is derived as the square root of the mean of squared per-task standard deviations.

To quantitatively evaluate the node classification results, we adopt accuracy and F1-score as standardized metrics. Accuracy is calculated as the ratio of correctly classified nodes to the total number of nodes, providing a simple measure of overall classification performance, given by:

$$\text{Accuracy} = \frac{\sum_{i=1}^{N} I(y_i = \hat{y}_i)}{N},$$

where $I(y_i = \hat{y}_i)$ is an indicator function that equals 1 if the true label $y_i$ matches the predicted label $\hat{y}_i$, and 0 otherwise, and $N$ is the total number of nodes. However, accuracy may not be sufficient when dealing with classification tasks. Therefore, we also use F1-score, which is the harmonic mean of precision and recall. Precision measures the proportion of true positive predictions among all positive predictions, while recall evaluates the proportion of true positives identified from all actual positives. The F1-score balances both metrics, offering a more reliable performance measure. Given the confusion-matrix counts (TP, FP, TN, FN) Where TP is the number of true positives, FP is the number of false positives, TN is the number of true negatives, and FN is the number of false negatives. The F1-score for each binary task is calculated as:

$$\text{Precision} = \frac{\text{TP}}{\text{TP} + \text{FP}}, \quad \text{Recall} = \frac{\text{TP}}{\text{TP} + \text{FN}}, \quad \text{F1-score} = 2 \times \frac{\text{Precision} \times \text{Recall}}{\text{Precision} + \text{Recall}}.$$

In multiclass settings, the micro-averaged F1-score aggregates counts over all classes when computing precision and recall. Let $\text{TP}_k$, $\text{FP}_k$, and $\text{FN}_k$ denote the true positives, false positives, and false negatives for class $k$. Define

$$\text{Precision}_{\text{micro}} = \frac{\sum_k \text{TP}_k}{\sum_k (\text{TP}_k + \text{FP}_k)}, \qquad \text{Recall}_{\text{micro}} = \frac{\sum_k \text{TP}_k}{\sum_k (\text{TP}_k + \text{FN}_k)},$$

and

$$\text{F1-score}_{\text{micro}} = \frac{2\,\text{Precision}_{\text{micro}}\,\text{Recall}_{\text{micro}}}{\text{Precision}_{\text{micro}} + \text{Recall}_{\text{micro}}}.$$

Micro-F1 reflects overall, sample frequency weighted performance. In this paper, we use micro-F1 for all F1-score evaluations.

For the calculation of accuracy in practice, we also set the maximum output token length limited to 100. We count a prediction as correct if the ground-truth label appears in the generated output, and report the proportion of correct predictions over the total number of samples. For F1, we extract a predicted label by scanning the output for the first match (falling back to unknown if none is found), then compute micro-F1 over multiple classes based on predicted labels and ground truths. We report the mean accuracy and F1-score of our method across three runs.

### A.3 BASELINES

**Baselines for Graph-level Classification**  We incorporate both non-LLM-based models and LLM-based models as baselines for graph classification tasks. The non-LLM-based models include graph prompt learning method GPF (Fang et al., 2023) with AttrMasking (Hu et al., 2020b) and GCL (You et al., 2020) as pretrained GNN models, and graph-language tuning method MolCA (Liu et al., 2023b) and its variant MolCA-GS. For the LLM-based models, we categorize the approaches as follows.

- Zero-shot Inference: Directly input textual graph attributes and instruction prompts into LLMs, utilizing their pre-trained knowledge and reasoning capabilities (Hu et al., 2024).
- Few-shot Learning: Directly input textual graph attributes, instruction prompts as well as examples with complete textual attributes and ground truth labels into LLMs, leveraging LLMs' in context learning capabilities (Coda-Forno et al., 2023), which means the few shot examples can guide LLMs' understanding of the task. The few shot examples sampling strategy include random sampling and selecting similar samples based on Morgan similarity (Morgan, 1965). In our settings, we use 8-shot learning (Guo et al., 2023) with 4 samples for each category.
- Supervised Fine-Tuning: Apply supervised fine-tuning via Low-Rank Adaptation (Hu et al., 2022) to adapt the model to downstream tasks. In our setup, we combine data from all tasks to form the training set, and use the jointly trained LLM to evaluate the test split of each task.
- Soft-Prompt: Employ a projector, i.e.a two-layer Multi-Layer Perceptron (MLP) with ReLU activation, to map graph embeddings into the LLM space, training only the MLP while keeping the LLM frozen. We use CrossEntropyLoss by comparing the LLM's predicted logit for the token "True" with the true labels (0/1), applied to the first generated token.

**Baselines for Node-level Classification**  We evaluate node classification against three baseline families: (i) graph self-supervised learning (SSL) methods that learn transferable structural representations (Li et al., 2023), exemplified by DGI (Velickovic et al., 2019) and GraphMAE (Hou et al., 2022); (ii) LLM-only baselines that rely solely on semantic information, like LLaMA3-70B (Dubey et al., 2024), GPT-3.5-turbo (Ouyang et al., 2022), GPT-4o (Hurst et al., 2024), and DeepSeek-chat (Guo et al., 2025); (iii) LLM–graph integration frameworks that leverage structural and semantic information within LLM-based pipelines for node classification. Emb w/ NA is a simple training-free baseline that combines structural and semantic cues for zero-shot inference (Li et al., 2024c). We also include two advanced integration frameworks: OFA (Liu et al., 2024), which textualizes all nodes and edges into a human-readable form, encodes heterogeneous domains into a shared language space, and adapts to downstream tasks by inserting task-specific prompting substructures; and ZeroG (Li et al., 2024b), which encodes node attributes and class descriptions with a language model and employs prompt-based subgraph sampling with lightweight fine-tuning to tackle cross-dataset zero-shot transfer. Collectively, these baselines span structure-only, semantics-only, and structure–semantics joint paradigms.

Some baseline results are taken from other works. In Table 1, the AUC-ROC results of GPT-4 come from Zhao et al. (2025); the results of GPT-4o, Deepseek-R1, and LLaMA3-70B come from Li et al. (2025); the results of Galactica-120B come from Liu et al. (2023a); the results of GIMLET come from Zhao et al. (2023a); the results of GPF-AttrMasking, GPF-GCL, MolCA-S, MolCA-GS come from Wang et al. (2025a). In Table 3, the accuracy and F1-score results of all baselines are all taken from Li et al. (2024c).

## B ADDITIONAL DETAILS ON IMPLEMENTATION

### B.1 MODEL DETAILS AND TRAINING HYPERPARAMETERS

We encode newly assigned structural attribute with SentenceTransformers all-MiniLM-L6-v2 (Reimers & Gurevych, 2019) for initial node features. We employ a two-layer GCN as the encoder in structural tokenizer, and set the size of structural vocabulary to 256. We adopt two widely used large language models LLaMA2-7B-chat (Touvron et al., 2023) and Llama-3.2-3B-Instruct (Dubey et al., 2024) as our backbones. All models are implemented in PyTorch (Paszke et al., 2019). In the structural tokenizer training phase, we firstly pretrain the GCN on reconstruction task as a warm-up with a learning rate of $1 \times 10^{-2}$. Then we optimize the whole structural tokenizer with two parameter groups: GCN with learning rate $5 \times 10^{-2}$ and structural tokens discrete embeddings 0.5, using

Adam optimizer (Kingma & Ba, 2015). In token alignment with hybrid QAs stage and task specific finetuning stage, we optimize the newly added token embeddings and LLM backbone with learning rate $5 \times 10^{-4}$ and a LoRA rank of 16, using AdamW optimizer (Loshchilov & Hutter, 2019).

## B.2 DETAILS OF DOWNSTREAM APPLICATIONS

After token alignment with hybrid QAs, we finetune the embeddings corresponding to the newly added structural tokens and the parameters of LLM backbone for application in downstream tasks. Due to severe class imbalance observed in the training sets of most tasks (shown in Table 6), we apply various data balancing techniques during finetuning. For some tasks, we also employ joint training with other tasks to finetune the LLM. Specifically, for the LLaMA3-3B model, we apply class balancing with a 1:1 ratio by duplicating minority class samples to match the majority on tasks BBBP_p_np, ClinTox_FDA_APPROVED, ClinTox_CT_TOX, and HIV_HIV_active, while retaining the original training data distribution for BACE_Class. Each task is finetuned separately, and evaluation is conducted using the checkpoint from the third epoch. For the Tox21 benchmark, we jointly finetune on the raw training data from all 17 tasks, including data from other four benchmarks, and evaluate the second-epoch checkpoint for Tox21 performance on its 12 subtasks. For the LLaMA2-7B model, we use unbalanced raw training data for Tox21_NR-ER and Tox21_SR-HSE, and apply 1:1 balancing ratio as mentioned above for Tox21_NR-AR-LBD, Tox2_NR-ER-LBD, Tox21_SR-ARE, ClinTox_FDA_APPROVED, ClinTox_CT_TOX, and HIV_HIV_active. For BBBP_p_np, Tox21_NR-AR, Tox21_NR-AhR, and Tox21_NR-PPAR-gamma, we resample the data such that the number of minority class samples is adjusted to one-fifth of the majority class samples, and for BACE_Class, we retain the original distribution. All tasks are finetuned individually, with evaluation based on the second epoch checkpoint, except for Tox21_NR-ER-LBD, which uses the third epoch. For the remaining Tox21 tasks (NR-Aromatase, SR-ATAD5, SR-MMP, SR-p53), we jointly finetune on unbalanced raw training data from all 17 tasks, evaluating the third epoch checkpoint for the four tasks. It's worth noting that we do not make any changes to the validation set and test set distribution.

## B.3 EXTENTION TO NODE-LEVEL CLASSIFICATION

To extend our method to node-level tasks, we construct ego-graphs by sampling neighborhood structure for each target node, enabling processing within the standard graph paradigm. For each node, we sample its 2-hop ego-graph, and map it to node-level structural tokens with our topology-aware structural tokenizer trained on graph tasks. We do not need to add a global node to the constructed ego-graphs any more. The specific $<\text{SO}\mathcal{G}_k>$ corresponding to the target node is chosen as the structural token to represent the local neighborhood topology of the target node and we insert it in the textual LLM input for the classification of the downstream nodes.

## B.4 PROMPT DESIGN OF $<\text{SO}\mathcal{G}_k>$

We incorporate structural tokens into the task prompt and textual graph attributes to construct complete LLM input prompt so that we can obtain class labels via generative inference. Detailed LLM input prompts for each task are provided in the corresponding boxes (from page 21 to 26).

## C ADDITIONAL CASE STUDIES

We provide additional examples illustrating how different structural tokens influence LLM's output probabilities discussed in Section 5.3, as shown in Figure 7, Figure 8, Figure 9. Also, we examine more examples of the selection of structural tokens and their corresponding structures in the training split of datasets (instead of test split in Section 5.4), shown in Figure 10, Figure 11.

## D USE OF LLMS

The core research presented in this paper directly incorporates Large Language Model as an integral component of our proposed pipeline, since we are dedicated to improving LLM's graph reasoning capabilities and mitigating structural hallucinations. Specifically, we add special structural tokens to the LLM input space for a more accurate and concise LLM topology understanding. The LLM serves as the main inference model to generate answers in our graph-benchmark evaluations, including graph classification and node classification tasks.

It is crucial to clarify that the LLM is used in this operational role and is not involved in the ideation, conceptualization, or writing of this paper. The research ideas and writing are entirely our own.

**LLM input prompts for BBBP_p_np**

<|begin_of_text|> <|start_header_id|> system<|end_header_id|>
You are a helpful and reliable assistant.<|eot_id|>
<|begin_of_text|><|start_header_id|> user<|end_header_id|>
You are a chemistry expert. Classify the given molecule into the correct category based on its molecular structure (token) and SMILES expression. Each structure token represents a unique graph pattern (e.g., a kind of similar molecular graphs).
[Molecule] $smiles\_expression\_of\_target\_molecule$
[Structure Token] $<SO\mathcal{G}_k>$
[Task] Does this molecule have blood-brain barrier permeability (BBB penetration)? True for BBB permeable and False for not BBB permeable. Output the complete correct answer from the following two options:
1. True
2. False
[Answer]<|eot_id|><|start_header_id|>LLM Assistant<|end_header_id|>

**LLM input prompts for Tox21_NR-AR**

<|begin_of_text|> <|start_header_id|> system<|end_header_id|>
You are a helpful and reliable assistant.<|eot_id|>
<|begin_of_text|><|start_header_id|> user<|end_header_id|>
You are a chemistry expert. Classify the given molecule into the correct category based on its molecular structure (token) and smiles expression. Each structure token represents a unique graph pattern (e.g., a kind of similar molecular graphs).
[Molecule] $smiles\_expression\_of\_target\_molecule$
[Structure Token] $<SO\mathcal{G}_k>$
[Task] Does this molecule activate the androgen receptor (NR-AR)? True for active and False for inactive. Output the complete correct answer from the following two options:
1. True
2. False
[Answer]<|eot_id|><|start_header_id|>LLM Assistant<|end_header_id|>

**LLM input prompts for Tox21_NR-AR-LBD**

<|begin_of_text|> <|start_header_id|> system<|end_header_id|>
You are a helpful and reliable assistant.<|eot_id|>
<|begin_of_text|><|start_header_id|> user<|end_header_id|>
You are a chemistry expert. Classify the given molecule into the correct category based on its molecular structure (token) and smiles expression. Each structure token represents a unique graph pattern (e.g., a kind of similar molecular graphs).
[Molecule] $smiles\_expression\_of\_target\_molecule$
[Structure Token] $<SO\mathcal{G}_k>$
[Task] Does this molecule activate the ligand-binding domain of the androgen receptor (NR-AR-LBD)? True for active and False for inactive. Output the complete correct answer from the following two options:
1. True
2. False
[Answer]<|eot_id|><|start_header_id|>LLM Assistant<|end_header_id|>

**LLM input prompts for Tox21_NR-AhR**

<|begin_of_text|> <|start_header_id|> system<|end_header_id|>
You are a helpful and reliable assistant.<|eot_id|>
<|begin_of_text|><|start_header_id|> user<|end_header_id|>
You are a chemistry expert. Classify the given molecule into the correct category based on its molecular structure (token) and smiles expression. Each structure token represents a unique graph pattern (e.g., a kind of similar molecular graphs).
[Molecule] *smiles_expression_of_target_molecule*
[Structure Token] <SO$\mathcal{G}_k$>
[Task] Does this molecule activate the aryl hydrocarbon receptor (NR-AhR)? True for active and False for inactive. Output the complete correct answer from the following two options:
1. True
2. False
[Answer]<|eot_id|><|start_header_id|>LLM Assistant<|end_header_id|>

**LLM input prompts for Tox21_NR-Aromatase**

<|begin_of_text|> <|start_header_id|> system<|end_header_id|>
You are a helpful and reliable assistant.<|eot_id|>
<|begin_of_text|><|start_header_id|> user<|end_header_id|>
You are a chemistry expert. Classify the given molecule into the correct category based on its molecular structure (token) and smiles expression. Each structure token represents a unique graph pattern (e.g., a kind of similar molecular graphs).
[Molecule] *smiles_expression_of_target_molecule*
[Structure Token] <SO$\mathcal{G}_k$>
[Task] Does this molecule inhibit the aromatase enzyme (NR-Aromatase)? True for active and False for inactive. Output the complete correct answer from the following two options:
1. True
2. False
[Answer]<|eot_id|><|start_header_id|>LLM Assistant<|end_header_id|>

**LLM input prompts for Tox21_NR-ER**

<|begin_of_text|> <|start_header_id|> system<|end_header_id|>
You are a helpful and reliable assistant.<|eot_id|>
<|begin_of_text|><|start_header_id|> user<|end_header_id|>
You are a chemistry expert. Classify the given molecule into the correct category based on its molecular structure (token) and smiles expression. Each structure token represents a unique graph pattern (e.g., a kind of similar molecular graphs).
[Molecule] *smiles_expression_of_target_molecule*
[Structure Token] <SO$\mathcal{G}_k$>
[Task] Does this molecule activate the estrogen receptor (NR-ER)? True for active and False for inactive. Output the complete correct answer from the following two options:
1. True
2. False
[Answer]<|eot_id|><|start_header_id|>LLM Assistant<|end_header_id|>

**LLM input prompts for Tox21_NR-ER-LBD**

<|begin_of_text|> <|start_header_id|> system<|end_header_id|>
You are a helpful and reliable assistant.<|eot_id|>
<|begin_of_text|><|start_header_id|> user<|end_header_id|>
You are a chemistry expert. Classify the given molecule into the correct category based on its molecular structure (token) and smiles expression. Each structure token represents a unique graph pattern (e.g., a kind of similar molecular graphs).
[Molecule] *smiles_expression_of_target_molecule*
[Structure Token] <SO$\mathcal{G}_k$>
[Task] Does this molecule activate the ligand-binding domain of the estrogen receptor (NR-ER-LBD)? True for active and False for inactive. Output the complete correct answer from the following two options:
1. True
2. False
[Answer]<|eot_id|><|start_header_id|>LLM Assistant<|end_header_id|>

**LLM input prompts for Tox21_NR-PPAR-gamma**

<|begin_of_text|> <|start_header_id|> system<|end_header_id|>
You are a helpful and reliable assistant.<|eot_id|>
<|begin_of_text|><|start_header_id|> user<|end_header_id|>
You are a chemistry expert. Classify the given molecule into the correct category based on its molecular structure (token) and smiles expression. Each structure token represents a unique graph pattern (e.g., a kind of similar molecular graphs).
[Molecule] *smiles_expression_of_target_molecule*
[Structure Token] <SO$\mathcal{G}_k$>
[Task] Does this molecule activate the peroxisome proliferator-activated receptor gamma (NR-PPAR-gamma)? True for active and False for inactive. Output the complete correct answer from the following two options:
1. True
2. False
[Answer]<|eot_id|><|start_header_id|>LLM Assistant<|end_header_id|>

**LLM input prompts for Tox21_SR-ARE**

<|begin_of_text|> <|start_header_id|> system<|end_header_id|>
You are a helpful and reliable assistant.<|eot_id|>
<|begin_of_text|><|start_header_id|> user<|end_header_id|>
You are a chemistry expert. Classify the given molecule into the correct category based on its molecular structure (token) and smiles expression. Each structure token represents a unique graph pattern (e.g., a kind of similar molecular graphs).
[Molecule] *smiles_expression_of_target_molecule*
[Structure Token] <SO$\mathcal{G}_k$>
[Task] Does this molecule activate the antioxidant response element (SR-ARE)? True for active and False for inactive. Output the complete correct answer from the following two options:
1. True
2. False
[Answer]<|eot_id|><|start_header_id|>LLM Assistant<|end_header_id|>

---

**LLM input prompts for Tox21_SR-ATAD5**

```
<|begin_of_text|> <|start_header_id|> system<|end_header_id|>
```
You are a helpful and reliable assistant.`<|eot_id|>`
```
<|begin_of_text|><|start_header_id|> user<|end_header_id|>
```
You are a chemistry expert. Classify the given molecule into the correct category based on its molecular structure (token) and smiles expression. Each structure token represents a unique graph pattern (e.g., a kind of similar molecular graphs).
[Molecule] *smiles_expression_of_target_molecule*
[Structure Token] `<SO`$\mathcal{G}_k$`>`
[Task] Does this molecule activate ATAD5 signaling (SR-ATAD5)? True for active and False for inactive. Output the complete correct answer from the following two options:
1. True
2. False
[Answer]`<|eot_id|><|start_header_id|>`LLM Assistant`<|end_header_id|>`

---

**LLM input prompts for Tox21_SR-HSE**

```
<|begin_of_text|> <|start_header_id|> system<|end_header_id|>
```
You are a helpful and reliable assistant.`<|eot_id|>`
```
<|begin_of_text|><|start_header_id|> user<|end_header_id|>
```
You are a chemistry expert. Classify the given molecule into the correct category based on its molecular structure (token) and smiles expression. Each structure token represents a unique graph pattern (e.g., a kind of similar molecular graphs).
[Molecule] *smiles_expression_of_target_molecule*
[Structure Token] `<SO`$\mathcal{G}_k$`>`
[Task] Does this molecule activate the heat shock element (SR-HSE)? True for active and False for inactive. Output the complete correct answer from the following two options:
1. True
2. False
[Answer]`<|eot_id|><|start_header_id|>`LLM Assistant`<|end_header_id|>`

---

**LLM input prompts for Tox21_SR-MMP**

```
<|begin_of_text|> <|start_header_id|> system<|end_header_id|>
```
You are a helpful and reliable assistant.`<|eot_id|>`
```
<|begin_of_text|><|start_header_id|> user<|end_header_id|>
```
You are a chemistry expert. Classify the given molecule into the correct category based on its molecular structure (token) and smiles expression. Each structure token represents a unique graph pattern (e.g., a kind of similar molecular graphs).
[Molecule] *smiles_expression_of_target_molecule*
[Structure Token] `<SO`$\mathcal{G}_k$`>`
[Task] Does this molecule activate the mitochondrial membrane potential stress response (SR-MMP)? True for active and False for inactive. Output the complete correct answer from the following two options:
1. True
2. False
[Answer]`<|eot_id|><|start_header_id|>`LLM Assistant`<|end_header_id|>`

**LLM input prompts for Tox21_SR-p53**

<|begin_of_text|> <|start_header_id|> system<|end_header_id|>
You are a helpful and reliable assistant.<|eot_id|>
<|begin_of_text|><|start_header_id|> user<|end_header_id|>
You are a chemistry expert. Classify the given molecule into the correct category based on its molecular structure (token) and smiles expression. Each structure token represents a unique graph pattern (e.g., a kind of similar molecular graphs).
[Molecule] *smiles_expression_of_target_molecule*
[Structure Token] <SO$\mathcal{G}_k$>
[Task] Does this molecule activate the p53 stress response pathway (SR-p53)? True for active and False for inactive. Output the complete correct answer from the following two options:
1. True
2. False
[Answer]<|eot_id|><|start_header_id|>LLM Assistant<|end_header_id|>

**LLM input prompts for ClinTox_FDA_APPROVED**

<|begin_of_text|> <|start_header_id|> system<|end_header_id|>
You are a helpful and reliable assistant.<|eot_id|>
<|begin_of_text|><|start_header_id|> user<|end_header_id|>
You are a chemistry expert. Classify the given molecule into the correct category based on its molecular structure (token) and smiles expression. Each structure token represents a unique graph pattern (e.g., a kind of similar molecular graphs).
[Molecule] *smiles_expression_of_target_molecule*
[Structure Token] <SO$\mathcal{G}_k$>
[Task] Has this molecule been approved by the FDA? True for FDA approved and false for non-approved. Output the complete correct answer from the following two options:
1. True
2. False
[Answer]<|eot_id|><|start_header_id|>LLM Assistant<|end_header_id|>

**LLM input prompts for ClinTox_CT_TOX**

<|begin_of_text|> <|start_header_id|> system<|end_header_id|>
You are a helpful and reliable assistant.<|eot_id|>
<|begin_of_text|><|start_header_id|> user<|end_header_id|>
You are a chemistry expert. Classify the given molecule into the correct category based on its molecular structure (token) and smiles expression. Each structure token represents a unique graph pattern (e.g., a kind of similar molecular graphs).
[Molecule] *smiles_expression_of_target_molecule*
[Structure Token] <SO$\mathcal{G}_k$>
[Task] Is this molecule associated with clinical toxicity? True for clinically toxic and false for non-toxic. Output the complete correct answer from the following two options:
1. True
2. False
[Answer]<|eot_id|><|start_header_id|>LLM Assistant<|end_header_id|>

---

**LLM input prompts for HIV_HIV_active**

<|begin_of_text|> <|start_header_id|> system<|end_header_id|>
You are a helpful and reliable assistant.<|eot_id|>
<|begin_of_text|><|start_header_id|> user<|end_header_id|>
You are a chemistry expert. Classify the given molecule into the correct category based on its molecular structure (token) and smiles expression. Each structure token represents a unique graph pattern (e.g., a kind of similar molecular graphs).
[Molecule] *smiles_expression_of_target_molecule*
[Structure Token] <SO$\mathcal{G}_k$>
[Task] Does this molecule inhibit HIV replication? True for active molecules which can inhibit HIV and false for inactive ones. Output the complete correct answer from the following two options:
1. True
2. False
[Answer]<|eot_id|><|start_header_id|>LLM Assistant<|end_header_id|>

---

**LLM input prompts for BACE_Class**

<|begin_of_text|> <|start_header_id|> system<|end_header_id|>
You are a helpful and reliable assistant.<|eot_id|>
<|begin_of_text|><|start_header_id|> user<|end_header_id|>
You are a chemistry expert. Classify the given molecule into the correct category based on its molecular structure (token) and smiles expression. Each structure token represents a unique graph pattern (e.g., a kind of similar molecular graphs).
[Molecule] *smiles_expression_of_target_molecule*
[Structure Token] <SO$\mathcal{G}_k$>
[Task] Is the binding result of the molecular on beta-secretase 1 true or false? Output the complete correct answer from the following two options:
1. True
2. False
[Answer]<|eot_id|><|start_header_id|>LLM Assistant<|end_header_id|>

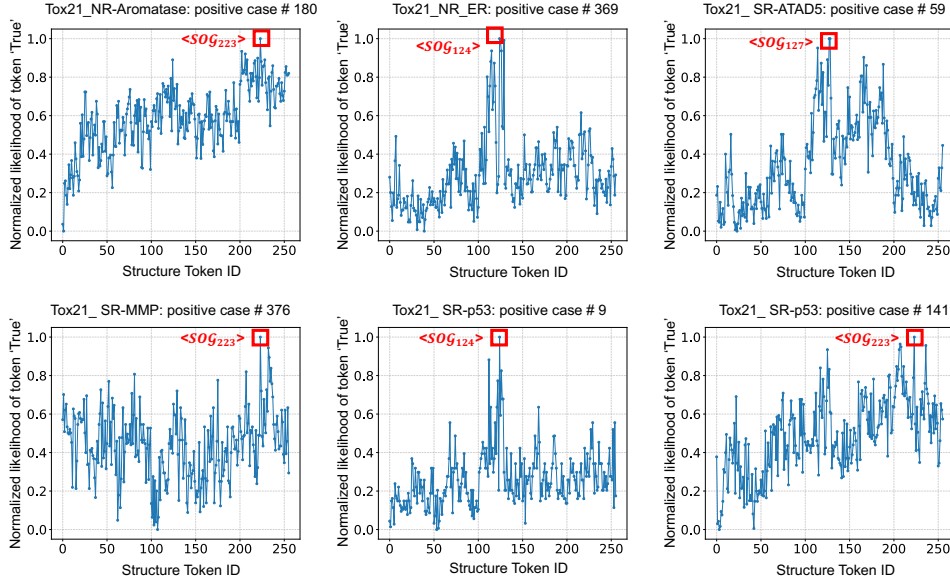

Figure 7: Example of performance variation across the selection of different structural tokens in Tox21 dataset.

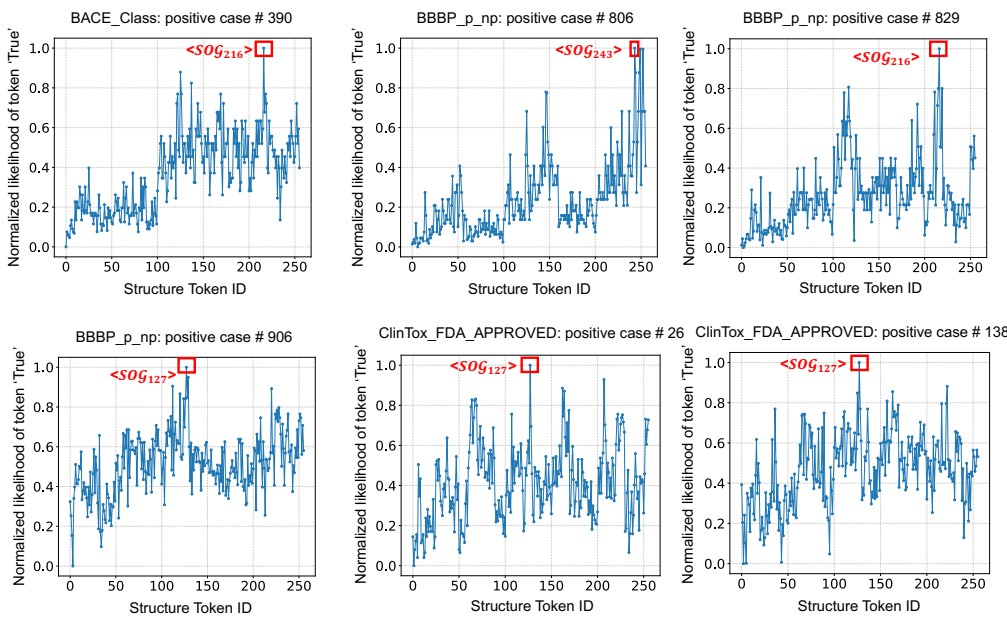

Figure 8: Example of performance variation across the selection of different structural tokens in BBBP, BACE and ClinTox dataset.

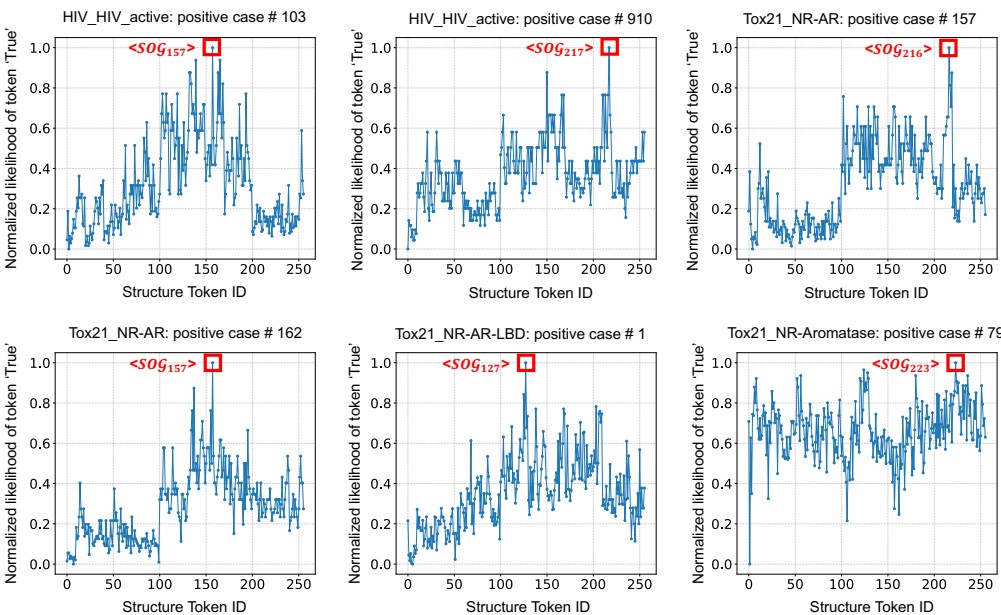

Figure 9: Example of performance variation across the selection of different structural tokens in Tox21 and HIV dataset.

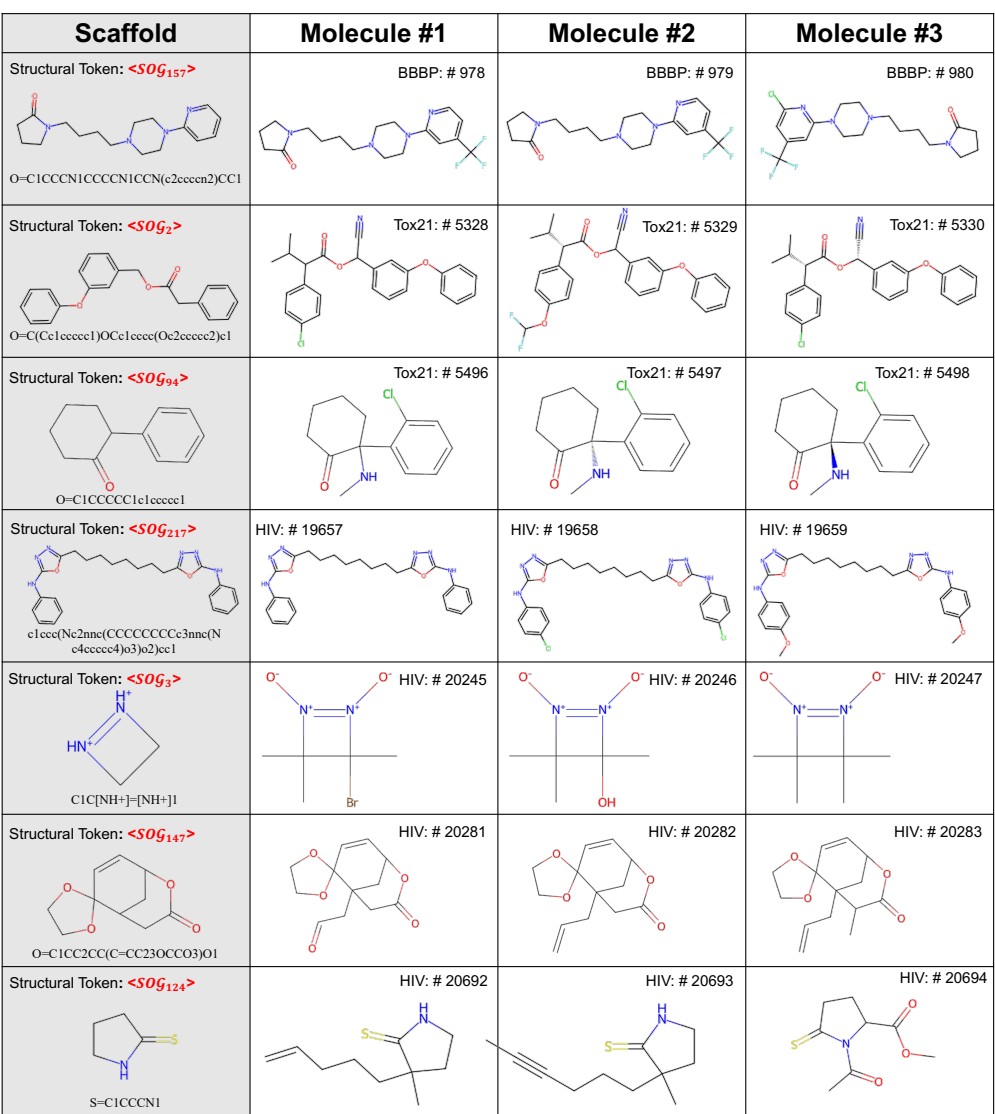

Figure 10: Molecules in same training set sharing the same scaffold with same structural token.

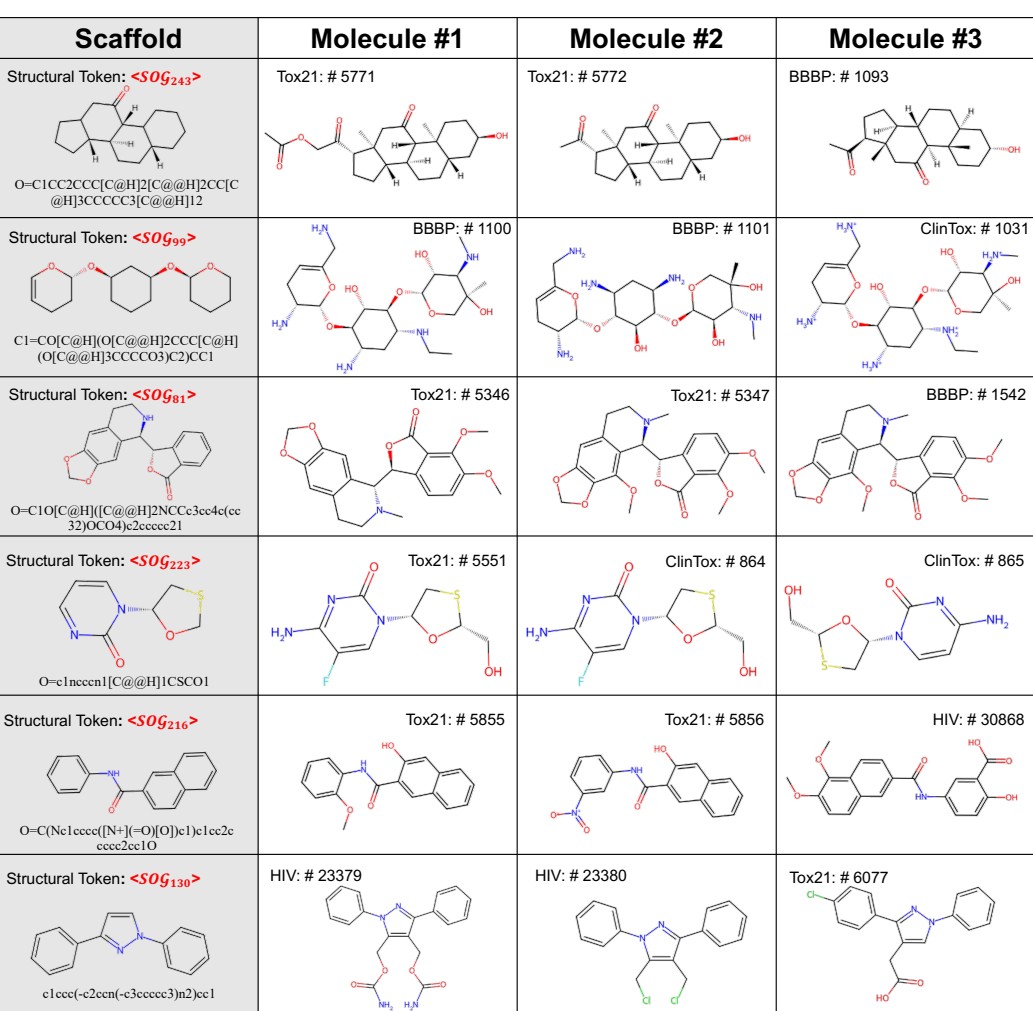

Figure 11: Molecules in different training sets sharing the same scaffold with same structural token.

