# OpenReview forum: "<SO$G_k$>: One LLM Token for Explicit Graph Structural Understanding"
_ICLR.cc/2026/Conference — ICLR 2026 Poster_

### Official Review · Reviewer_g287 · 2025-10-25

**Soundness:** 3
**Presentation:** 2
**Contribution:** 2
**Rating:** 6
**Confidence:** 3

**Summary:**

To address the structural hallucination problem of LLMs)on graph-structured data—where existing Graph-to-Text approaches suffer from high token consumption and attention dilution, and Graph-to-Embedding methods face cross-modal misalignment—this work proposes a novel solution: introducing a **single special token** `<SOGₖ>` . A topology-aware graph tokenizer maps the entire graph topology into a highly selective discrete token from a learnable structural vocabulary of size K, which is then injected into the LLM input sequence, forming a hybrid architecture that blends structural and textual modalities.

**Strengths:**

- **Innovative and elegant design**: Representing an entire graph via a single virtual global node mapped to a vocabulary index is conceptually clean and computationally efficient.
- **Unified representation space**: By training on hybrid QA corpora (including k-NN matching, true/false judgment, and description alignment tasks), `<SOGₖ>` is embedded in the **same semantic space** as native LLM text tokens, effectively eliminating the modality gap inherent in Graph-to-Embedding methods.
- **High efficiency and strong performance**: The approach achieves competitive or superior results with minimal token overhead (just +1 token) and negligible parameter cost.
- **Interpretability**: The paper includes meaningful analysis showing that each `<SOGₖ>` token occupies a distinct region in the embedding space, suggesting meaningful structural encoding.

**Weaknesses:**

- **Insufficient related work discussion**: To the best of my knowledge, similar ideas have appeared in prior literature(1) *Multi-View Empowered Structural Graph Wordification for Language Models*, which also uses discrete tokens to represent global graph structure; (2) *LangTopo*
  The paper does not adequately differentiate itself from these works.
- **Lack of ablation studies**: The proposed “topology-aware graph tokenizer” comprises three key components—(i) anchor node selection (via node degree), (ii) virtual global node , and (iii) self-supervised topological reconstruction—but none are ablated independently to assess their individual contributions.
- **Missing hyperparameter sensitivity analysis**: The structural vocabulary size K is fixed at 256 without exploring alternatives (e.g., K=64, 128, 512). It remains unclear whether performance saturates, degrades, or improves with larger/smaller K, raising concerns about robustness and scalability.

**Questions:**

1. **Multi-token extension**: The single-token design is indeed attractive. However, has the authors explored using **multiple structural tokens** (e.g., 2–5 tokens) to capture richer or hierarchical graph properties? Could this further improve performance on complex graphs?

2. **Relation to prior work**: How does this method fundamentally differ from *LangTopo* and *Dr.E*? Is the key novelty the **end-to-end joint training with LLMs in a shared token space**, or the **specific topology-aware tokenizer design**? Clarifying this distinction would strengthen the paper’s positioning.

---

> ### Author Response · Authors · 2025-11-25
>
> # Response to g287 (1/4)
>
> We sincerely appreciate your support for our work, particularly your recognition of our core idea of using one single token to represent topologies, which you have described as innovative and elegant. Additionally, we are exited that you agree with aligning those structural tokens through hybrid QAs. You also noted that <SOGₖ> is efficient, strong and interpretable, which highlights the advantages inherent in our design. It is a great privilege to have encountered a reviewer who so thoroughly understands our ideas. Your feedback is not only encouraging but also significant to us. Especially given the challenging nature of <SOGₖ>, we deeply appreciate how accurately you have identified and pointed out its potential. We sincerely hope that you will continue to support our work.
>
> #### **Q1:**
>
> We are so glad to see that you think our **single-token design** is attractive. It is also our main research goal to incorporate **only one token** and we've demonstrated it has great potential for effective topology understanding, as shown in Sec 5. Actually, the critical substructures responsible for decision-making typically constitute only a small fraction of the entire graph [1,2], which supports our ​**one-token goal**​.
>
> Motivated by your suggestion, we further conduct experiments extending to multiple structural tokens (e.g., 2–5 tokens), which is practical on complex graphs in real applications by graph partitioning and community detection,etc. To keep things simple, we extend to top-2, top-3, and top-5 structural tokens.
>
> | LLaMA3.2-3B |                        |       |          |          |          |
> | ------------- | ------------------------ | ------- | ---------- | ---------- | ---------- |
> | Dataset     | Task                   | SOGₖ | 2 tokens | 3 tokens | 5 tokens |
> | ClinTox     | ClinTox\_FDA\_APPROVED | 76.1  | 74.1     | 74.1     | 64.8     |
> |             | ClinTox\_CT\_TOX       | 95.0  | 98.3     | 96.7     | 96.7     |
> | HIV         | HIV\_HIV\_active       | 75.7  | 56.5     | 56.3     | 56.4     |
> | BACE        | BACE\_Class            | 63.3  | 72.9     | 73.4     | 70.7     |
> | BBBP        | BBBP\_p\_np            | 76.9  | 81.5     | 81.3     | 81.2     |
> |             | avg.                   | __77.4__  | 76.7     | 76.4     | 74.0     |
>
> | LLaMA2-7B |                        |       |          |          |          |
> | ----------- | ------------------------ | ------- | ---------- | ---------- | ---------- |
> | Dataset   | Task                   | SOGₖ | 2 tokens | 3 tokens | 5 tokens |
> | ClinTox   | ClinTox\_FDA\_APPROVED | 94.1  | 92.2     | 95.9     | 94.1     |
> |           | ClinTox\_CT\_TOX       | 94.5  | 63.2     | 66.7     | 66.7     |
> | HIV       | HIV\_HIV\_active       | 83.2  | 81.7     | 81.3     | 81.3     |
> | BACE      | BACE\_Class            | 98.4  | 91.7     | 93.2     | 93.3     |
> | Tox21     | Tox21\_NR-AR-LBD       | 81.6  | 79.8     | 73.7     | 76.3     |
> |           | Tox21\_NR-AR           | 83.3  | 70.7     | 72.8     | 75.3     |
> |           | Tox21\_NR-AhR          | 86.6  | 90.0     | 89.7     | 89.5     |
> |           | Tox21\_NR-ER           | 84.3  | 64.3     | 66.2     | 64.5     |
> |           | Tox21\_NR-ER-LBD       | 78.2  | 63.5     | 63.5     | 61.9     |
> |           | Tox21\_NR-PPAR-gamma   | 72.5  | 81.1     | 83.3     | 78.0     |
> |           | Tox21\_SR-ARE          | 70.3  | 80.1     | 77.7     | 76.2     |
> |           | Tox21\_SR-HSE          | 84.0  | 74.8     | 71.3     | 72.7     |
> |      |       avg.                 | __84.3__  | 77.8     | 77.9     | 77.5     |
>
> The results show that, in some cases, such as BBBP on LLaMA3-3B and Tox21\_NR-AhR,Tox21\_NR-PPAR-gamma,Tox21\_SR-ARE on LLaMA2-7B, using *more than one* token indeed yields improvements. As you kindly pointed out, more complex graphs naturally benefit from more tokens that can represent richer characteristics. However, in the majority of datasets, we find that the ​**single-token design is enough and performs best overall**​, offering a favorable balance between structural fidelity and token efficiency.
>
> [1] Yuan, Hao, et al. "On explainability of graph neural networks via subgraph explorations." *International conference on ​*​​*machine learning*​. PMLR, 2021.
>
> [2] Wu, Zhenxing, et al. "Chemistry-intuitive explanation of graph neural networks for molecular property prediction with substructure masking." *Nature Communications* 14.1 (2023): 2585.

---

> ### Author Response · Authors · 2025-11-25
>
> # Response to g287 (2/4)
>
> #### **Q2/W1: ​**
>
> Thank you for directing us to these two relevant and valuable recent works. **We have added them and our discussion in the Related Work section and highlighted all revised text in red.** And we are very interested in your insightful questions. We would like to respond to each point as follows.
>
> **(1) The main difference between <SOGₖ> and LangTopo is that we use a Graph-to-Tokens approach, while LangTopo follows the Graph-to-Text paradigm.**
>
> Our method proposes using one specialized token, <SOGₖ>, to represent each Structure Of Graph and directly input it into the LLM. In contrast, LangTopo leverages pure natural language as input for linkage information.
>
> While both approaches utilize quantization techniques to abstract structural information, we quantize it into a single token(codeword in LangTopo) as the LLM input, whereas LangTopo uses the quantized relaxed distribution and vectors as supervisory signals for LLM output.
>
> Our goal is to explicitly represent the graph structure as structural tokens, which are designed to adapt to the LLM’s input characteristics and facilitate structural information sharing. We are adopting a Graph-to-Token paradigm. On the other hand, LangTopo fits more within a Graph-to-Text paradigm, where the primary focus is on helping the LLM inherently interpret the graph structure through natural language.
>
> (2) **As for <SOGₖ> and "Multi-View Empowered Structural Graph Wordification for Language Models", namely Dr.E. , we differ both in ​**​***what***​**​ is tokenized and ​**​***why***​​**​ it is tokenized**​.
>
> Dr.E uses the LLM’s existing vocabulary as the quantization codebook to quantize multi-hop graph information. This design intentionally mixes structural and textual information together, leveraging existing LLM embeddings to represent mixed information. Intuitively, it is a high-level keywords compression and still uses natural language input, which is very explainable and cost-effective.
>
> By contrast, we seperate graph topologies out and introduce a completely new set of tokens dedicated to representing them, creating a distinct “language” for pure structural information. Our goal is to **explicitly** express the decoupled topology, so that structural tokens provide a more focused, interpretable, and reusable interface and facilitate structural information sharing.
>
> (3) **The key novelty of our work lies not in introducing any individual component but in the idea of incorporating one special token <SOGₖ> to fully represent the graph structure within a unified token space for LLMs.**
>
> Indeed, this concept is realized through a simple yet effective **specific topology-aware tokenizer** and hybridQA alignment (​**end-to-end joint training with LLMs**​) as you have mentioned. However, the core meaning is their help to demonstrate that one single token <SOGₖ> can empower LLMs to understand, generate, and reason about graph structures in a concise and accurate manner.
>
> #### **W2: ​**
>
> We sincerely appreciate your insightful advice. In response, we have added more experiments to further demonstrate the contributions of each component in topology-aware graph tokenizer. To be specific, we ablate three key components.**​ ​**
>
> **(i) anchor node selection**
>
> In the original manuscript, we use node **degree** to select anchor node and locate other nodes accordingly. We further compare AUC-ROC on benchmarks with more anchor-selection strategies to examine its influence on structural encoding and downstream performance.
>
> |                        | random | pagerank | betweeness | degree\* |
> | ------------------------ | -------- | ---------- | ------------ | ---------- |
> | BBBP\_p\_np            | 74.4   | 82.2     | 80.9       | 76.9     |
> | HIV\_HIV\_active       | 72.0   | 75.4     | 62.3       | 79.7     |
> | BACE\_Class            | 63.5   | 64.3     | 65.5       | 98.4     |
> | ClinTox\_FDA\_APPROVED | 92.4   | 99.6     | 98.0       | 77.8     |
> | Tox21\_NR-AR           | 69.6   | 59.6     | 69.7       | 76.4     |
> | Tox21\_NR-AR-LBD       | 66.6   | 58.7     | 64.0       | 76.0     |
> | Tox21\_SR-p53          | 66.6   | 54.6     | 56.4       | 63.3     |
>
> We find that in most cases, the simple and effective **degree** strategy remains the best choice. For BBBP and ClinTox, **PageRank** performs slightly better because it better captures global centrality under sparse molecular graphs. Conversely, in most settings, **random** anchor selection shifts the structural center of graphs unpredictably, leading to unstable structural encoding and degraded performance.

---

> > ### Author Response · Authors · 2025-11-25
> >
> > # Response to g287 (3/4)
> > **(ii) virtual global node**
> >
> > Our original design uses a **global node** to perform structural pooling. We eliminate the global node (use anchor node instead) to assess influences of structural information aggregation strategy.
> >
> > | Dataset | w/o global node | original |
> > | --------- | ----------------- | ---------- |
> > | HIV     | 52.8            | 79.7     |
> > | BACE    | 54.9            | 98.4     |
> > | Tox21   | 53.7            | 81.6     |
> >
> > We also provide AUC-ROC results for each task specifically.
> >
> > | Task                 | w/o global node | original |
> > | ---------------------- | ----------------- | ---------- |
> > | HIV\_HIV\_active     | 52.8            | 79.7     |
> > | BACE\_Class          | 54.9            | 98.4     |
> > | Tox21\_NR-AR         | 61.8            | 76.4     |
> > | Tox21\_NR-AR-LBD     | 57.6            | 76.0     |
> > | Tox21\_NR-AhR        | 58.1            | 94.1     |
> > | Tox21\_NR-Aromatase  | 50.5            | 82.3     |
> > | Tox21\_NR-ER         | 57.7            | 84.9     |
> > | Tox21\_NR-ER-LBD     | 51.9            | 87.8     |
> > | Tox21\_NR-PPAR-gamma | 52.3            | 82.8     |
> > | Tox21\_SR-ARE        | 51.6            | 85.2     |
> > | Tox21\_SR-ATAD5      | 51.0            | 76.1     |
> > | Tox21\_SR-HSE        | 49.8            | 95.0     |
> > | Tox21\_SR-MMP        | 51.7            | 75.7     |
> > | Tox21\_SR-p53        | 50.6            | 63.3     |
> >
> > We observe a significant degradation in AUC-ROC performance when the global node is removed, demonstrating its effectiveness in capturing and compressing essential structural information ​**in a concise and efficient manner**​.
> >
> > **(iii) self-supervised topological reconstruction**
> >
> > The tokenizer’s training loss contains a self-supervised reconstruction term. To assess its impact, we remove this loss (keeping only the warmup step) and evaluate the resulting performance.
> >
> > | Dataset | w/o reconstruction | original |
> > | --------- | -------------------- | ---------- |
> > | HIV     | 71.5               | 79.7     |
> > | BACE    | 66.1               | 98.4     |
> > | BBBP    | 78.9               | 76.9     |
> > | Tox21   | 54.0               | 81.6     |
> >
> > We also provide AUC-ROC results for each task specifically.
> >
> > | Task                 | w/o reconstruction | original |
> > | ---------------------- | -------------------- | ---------- |
> > | HIV\_HIV\_active     | 71.5               | 79.7     |
> > | BACE\_Class          | 66.1               | 98.4     |
> > | BBBP\_p\_np          | 78.9               | 76.9     |
> > | Tox21\_NR-AR         | 59.7               | 76.4     |
> > | Tox21\_NR-AR-LBD     | 62.1               | 76.0     |
> > | Tox21\_NR-AhR        | 60.4               | 94.1     |
> > | Tox21\_NR-Aromatase  | 50.0               | 82.3     |
> > | Tox21\_NR-ER         | 54.5               | 84.9     |
> > | Tox21\_NR-ER-LBD     | 53.8               | 87.8     |
> > | Tox21\_NR-PPAR-gamma | 52.3               | 82.8     |
> > | Tox21\_SR-ARE        | 50.6               | 85.2     |
> > | Tox21\_SR-ATAD5      | 50.5               | 76.1     |
> > | Tox21\_SR-HSE        | 51.1               | 95.0     |
> > | Tox21\_SR-MMP        | 52.8               | 75.7     |
> > | Tox21\_SR-p53        | 50.5               | 63.3     |
> >
> > After removing the reconstruction loss, the performance generally degrades, with a dramatic 27.6 AUC-ROC drop on Tox21. The absence of the self-supervised reconstruction component prevents the tokenizer from adequately capturing structural information and embedding it into the structural vocabulary.

---

> > > ### Author Response · Authors · 2025-11-25
> > >
> > > # Response to g287 (4/4)
> > > #### **W3: ​**
> > >
> > > We appreciate your valuable suggestion. Following your advice, we have now included a hyperparameter analysis, as presented below, evaluating the structural vocabulary size K in different values (e.g., K=64, 128, 256, 512).
> > >
> > > From the average results, we observe that **K = 256** yields the strongest overall performance. When *K* is too small, many structural patterns cannot be sufficiently distinguished; when *K* is too large, the model tends to overfit and loses generalization ability. **K = 256 ​**provides an effective balance.
> > >
> > > | Dataset | K = 64 | K = 128 | K = 256\* | K = 512 |
> > > | --------- | -------- | --------- | ----------- | --------- |
> > > | BBBP    | 84.6   | 71.8    | 76.9      | 96.2    |
> > > | ClinTox | 91.8   | 90.1    | 76.9      | 90.0      |
> > > | HIV     | 52.7   | 51.9    | 79.7      | 55.9    |
> > > | BACE    | 69.8   | 62.3    | 98.4      | 85.7    |
> > > | Tox21   | 52.6   | 53.4    | 81.6      | 59.8    |
> > > | avg.    | 60.1   | 59.3    | 81.7      | 66.8    |
> > >
> > > We also provide specific AUC-ROC results for each 17 tasks.
> > >
> > > | Task                   | K = 64      | K = 128     | K = 256\*   | K = 512      |
> > > | ------------------------ | ------------- | ------------- | ------------- | -------------- |
> > > | BBBP\_p\_np            | 84.6 ± 3.2 | 71.8 ± 3.4 | 76.9 ± 3.1 | 96.2 ± 0.8  |
> > > | ClinTox\_FDA\_APPROVED | 95.2 ± 2.9 | 92.5 ± 0.2 | 77.8 ± 4.1 | 100.0 ± 0.0 |
> > > | ClinTox\_CT\_TOX       | 88.5 ± 2.5 | 87.6 ± 3.8 | 76.1 ± 3.4 | 80.0 ± 10.0 |
> > > | HIV\_HIV\_active       | 52.7 ± 0.0 | 51.9 ± 0.4 | 79.7 ± 3.0 | 55.9 ± 2.0  |
> > > | BACE\_Class            | 69.8 ± 6.1 | 62.3 ± 2.1 | 98.4 ± 0.9 | 85.7 ± 1.7  |
> > > | Tox21\_NR-AR           | 60.7 ± 3.7 | 61.3 ± 2.2 | 76.4 ± 2.1 | 60.9 ± 1.8  |
> > > | Tox21\_NR-AR-LBD       | 58.5 ± 1.5 | 58.6 ± 6.5 | 76.0 ± 6.7 | 55.2 ± 2.6  |
> > > | Tox21\_NR-AhR          | 55.4 ± 0.3 | 56.2 ± 1.0 | 94.1 ± 2.9 | 61.8 ± 1.1  |
> > > | Tox21\_NR-Aromatase    | 50.0 ± 0.0 | 50.3 ± 0.6 | 82.3 ± 1.4 | 70.6 ± 2.2  |
> > > | Tox21\_NR-ER           | 51.6 ± 0.3 | 55.8 ± 1.6 | 84.9 ± 4.1 | 67.6 ± 2.6  |
> > > | Tox21\_NR-ER-LBD       | 52.2 ± 0.1 | 53.8 ± 1.4 | 87.8 ± 2.6 | 60.9 ± 2.7  |
> > > | Tox21\_NR-PPAR-gamma   | 50.0 ± 0.0 | 51.5 ± 1.3 | 82.8 ± 1.5 | 56.8 ± 2.3  |
> > > | Tox21\_SR-ARE          | 50.3 ± 1.3 | 50.6 ± 1.8 | 85.2 ± 2.6 | 57.5 ± 2.0  |
> > > | Tox21\_SR-ATAD5        | 49.9 ± 0.1 | 50.0 ± 0.0 | 76.1 ± 5.3 | 59.0 ± 3.1  |
> > > | Tox21\_SR-HSE          | 50.7 ± 0.6 | 50.0 ± 0.0 | 95.0 ± 0.1 | 52.1 ± 1.0  |
> > > | Tox21\_SR-MMP          | 51.7 ± 1.0 | 52.8 ± 2.2 | 75.7 ± 1.6 | 57.3 ± 0.4  |
> > > | Tox21\_SR-p53          | 50.0 ± 0.0 | 50.6 ± 0.6 | 63.3 ± 4.2 | 57.6 ± 1.8  |
> > > | __avg.__                   | __60.1__        | __59.3__        | __81.7__        | __66.8__         |
> > >
> > > Finally, we are grateful for your thoughtful feedback, and we hope our response has clarified your concerns and strengthened your trust. We are committed to continuing to improve and build upon these results, and we truly appreciate your support as we move forward.

---

### Official Review · Reviewer_tD9C · 2025-11-01

**Soundness:** 3
**Presentation:** 4
**Contribution:** 2
**Rating:** 4
**Confidence:** 3

**Summary:**

The paper introduces a method that maps each graph’s topology to a single discrete structural token <SOG_k> that can be directly fed into an LLM, avoiding long graph-to-text encodings.
It builds these tokens through a topology-aware tokenizer and aligns them with the LLM’s embedding space using structure-based QA fine-tuning.

**Strengths:**

- One discrete token injects topology into the LLM, drastically reducing prompt length vs graph-to-text.

- Large improvements on multiple MoleculeNet tasks with clear ablations.

- Interpretability analysis is insightful.

- Works with off-the-shelf LLMs (LoRA + token embeddings), no architectural changes to the base model.

**Weaknesses:**

- It remains unclear whether this method scales robustly to heterogeneous or large real-world graphs (e.g., knowledge graphs, citation networks, or synthetic structural datasets).

- The approach is not novel. Some papers [1,2,3,4] have already provided similar approaches. Moreover these baselines are missing.

- Permutation/anchor bias: Tokenizer choices (anchor selection, hop labeling) may introduce ordering biases; robustness to isomorphisms not rigorously tested.

- Not generalizable. Requires fine tuning for each new dataset.

[1] Perozzi, Bryan, et al. "Let your graph do the talking: Encoding structured data for llms." arXiv preprint arXiv:2402.05862 (2024).

[2] Runjin Chen, Tong Zhao, Ajay Jaiswal, Neil Shah, and Zhangyang Wang. Llaga: Large language and
graph assistant. arXiv preprint arXiv:2402.08170, 2024b.

[3] Chai, Ziwei, et al. "Graphllm: Boosting graph reasoning ability of large language model." arXiv preprint arXiv:2310.05845 (2023).

[4] Xiaoxin He, Yijun Tian, Yifei Sun, Nitesh V Chawla, Thomas Laurent, Yann LeCun, Xavier Bresson,
and Bryan Hooi. G-retriever: Retrieval-augmented generation for textual graph understanding and
question answering. arXiv preprint arXiv:2402.07630, 2024

**Questions:**

- Generalization: How does SOGk perform on non-molecular benchmarks (e.g., social/citation/synthetic graphs) and on node- or link- level tasks?

- Invariance: Which parts of the tokenizer are provably permutation-invariant? What breaks if node ordering or anchor selection is adversarially perturbed?

---

> ### Author Response · Authors · 2025-11-25
>
> # Response to tD9C (1/2)
>
> Thank you for your valuable feedback. We greatly appreciate your recognition of the core contributions of our method, the integration of a single structural token to represent graph topology, and the interpretability analyses as well as the simplicity of applying our method to off-the-shelf LLMs. At the same time, we hope our clarifications and additional experiments can illustrate some aspects you pointed out.
>
> #### ​**W1/Q1**​:
>
> Thank you for pointing out this important issue. Besides molecular graphs, **we have already presented our evaluations on citation networks Cora (2,485 nodes) and Pubmed (19,717 nodes) in Tab.3 ​**and achieved substantial improvements. Due to the time and computational resources, we try our best to supplement experiments on **OGBN-Arxiv (169,343 nodes,citation networks)** and **​SST-2 (tree-structured patterns,​​​sentiment analysis)**, ​comparing with publicly reported performance in related literature [1,2,3].**​ ​**The results demonstrate our scalability across diverse domains and larger graphs.
>
> | OGBN-Arxiv       |              |               |
> | ------------------ | -------------- | --------------- |
> | Technique        | Methods      | Accuracy      |
> | GNN-based        | GCN          | 72.24         |
> |                  | GAT          | 71.85         |
> |                  | GraphSAGE    | 71.88         |
> | LLM as enhancer  | GIANT        | 72.04         |
> |                  | TAPE         | 72.99         |
> |                  | OFA          | 73.23         |
> |                  | ENGINE       | 75.01         |
> | LLM as predictor | GraphAdapter | 74.45         |
> |                  | LLaGA        | 72.78         |
> |                  | InstructGLM  | 39.09         |
> |                  | GraphText    | 49.47         |
> |                  | SOGₖ        | __78.22__ |
>
> | SST-2     |                        |          |
> | ----------- | ------------------------ | ---------- |
> | Technique | Methods                | Accuracy |
> | GNN-based | ERM                    | 79.3     |
> |           | ASAP                   | 81.3     |
> |           | IRM                    | 81.1     |
> |           | Group DRO              | 81.1     |
> |           | GIL                    | 83.4     |
> |           | LiSA                   | 80.4     |
> | LLM-based | GPT2-base              | 50.2     |
> |           | GPT2-large             | 63.3     |
> |           | OPT-6.7B               | 93.5     |
> |           | OPT-13B                | 96.0       |
> |           | InstructGPT-3          | 91.6     |
> |           | InstructGPT-3-few-shot | 92.4     |
> |           | InstructGPT-3-CoT      | 92.1     |
> | LLaMA3-3B | LLaMA3-3B              | 65.7     |
> |           | LLaMA3-3B-few-shot     | 67.3     |
> |           | SOGₖ (LLaMA3-3B)      | __98.7__     |
>
> Due to time constraints, we performed downsampling during the LLM fine-tuning stage, training the model on only 2% of the original training data selected at random. For evaluation, we report results on a randomly sampled subset of 973 instances from the test set.
>
> [1] Li, Yuhan, et al. "Glbench: A comprehensive benchmark for graph with large language models." *Advances in Neural Information Processing Systems* 37 (2024): 42349-42368.
>
> [2] Tang, Ruixiang, Dehan Kong, and Longtao Huang. "Large language models can be lazy learners: Analyze shortcuts in in-context learning." ​*Findings of the Association for Computational Linguistics: ACL 2023*​. 2023.
>
> [3] Sun, Xiaofei, et al. "Text classification via large language models." *arXiv preprint arXiv:2305.08377* (2023).
>
> #### ​**W2**​:
>
> Thank you for your comment. We fully agree that these works represent strong progress in **graph-to-embedding** paradigm. **However, we are fundamentally different in technical route since we follow graph-to-token paradigm.** To answer the question, we respectfully clarify that the cited works fall into two different categories.
>
> ​**First**​, paper **[1]** (​*Let Your Graph Do the Talking*​) has already been cited and compared as a baseline. Specifically, its soft-prompt design corresponds to the *Soft Prompt* baseline in our experiments (Sec. 5.1, line 307).
>
> ​**Second**​, papers **[2] LLaGA**, **[3] GraphLLM**, and **[4] G-retriever** follow the ​**graph-to-embedding paradigm**​, where graph representations are projected into the continuous embedding space of an LLM. We have already analyzed this issue in the manuscript and included representative baselines from this category. As discussed in  **Sec. 2 (Graph-to-Embedding methods)** and shown empirically in ​**Fig. 6**​, **this technical route suffers from severe misalignment between the inherent token space of the LLM and the projected graph embeddings.**  And we take your advice and added [3,4] ([2] exists already) in the Related Work section under “Graph-to-Embedding methods”. We thank the reviewer again for this helpful suggestion.

---

> ### Author Response · Authors · 2025-11-25
>
> # Response to tD9C (2/2)
> #### ​**W3/Q2**​:
>
> Thank you for raising this insightful point. While some ordering bias may arise in the early tokenization steps, our hybrid-QAs alignment design progressively mitigates this effect, as detailed below.
>
> The anchor node is the core of node encoding determining node locations and thus initializations. To alleviate the potential biases from anchor node, in Section 4.2, we introduce hybrid-QAs alignment, particularly the Description-Token Pairs Matching**​ ​**QA task, which challenges the LLM to overcome the ordering bias and to perform fine-grained structural reasoning, by comprehensing specific nodes and edges and form overall patterns, linking them to the correct graph token in a position-agnostic manner. Ablation study of hybrid-QAs is in Sec 5.3.
>
> Motivated by your concern, we include additional experiments where we vary the anchor-selection strategy (degree, PageRank, betweenness centrality, and random).
>
> |                        | pagerank | betweeness | random | degree\* |
> | ------------------------ | ---------- | ------------ | -------- | ---------- |
> | BBBP\_p\_np            | __82.2__    | 80.9       | 74.4   | 76.9     |
> | ClinTox\_FDA\_APPROVED | __99.6__     | 98.0         | 92.4   | 77.8     |
> | HIV\_HIV\_active       | 75.4     | 62.3       | 72.0     | __79.7__     |
> | Tox21\_NR-AR           | 59.6     | 69.7       | 69.6   | __76.4__     |
> | Tox21\_NR-AR-LBD       | 58.7     | 64.0         | 66.6   | __76.0__       |
> | Tox21\_SR-p53          | 54.6     | 56.4       | __66.6__   | 63.3     |
> | Tox21\_SR-ATAD5        | 56.6     | 68.7       | 75.8   | __76.1__     |
> | Tox21\_SR-MMP          | 56.4     | 64.1       | 72.1   | __75.7__     |
> | Tox21\_SR-p53          | 54.6     | 56.4       | __66.6__   | 63.3     |
> | avg.                   | 66.4     | 68.9       | 72.9   | __73.9__     |
>
> As shown in the results, while individual datasets may favor different centrality measures, ​**the overall performance remains in a comparable range across these choices**​. This indicates that the tokenizer does not rely on a specific node ordering to function correctly.
>
> #### ​**W4**​:
>
> We sincerely thank you for highlighting this issue. We offer the following explanation about fine-tuning in our framework. We would like to clarify that the fine-tuning required by our method ​**is both lightweight and acceptable in practice**​, and it aligns structural tokens with domain-specific textual semantics in vertical domains. This alignment step enhances the LLM’s ability to jointly interpret graph structure and text in settings such as molecules or citation networks.
>
> Importantly, the method ​**does not rely on fine-tuning to function**​. Structural tokens already provide observable gains even before any domain adaptation, as shown in our pre–fine-tuning results.
>
> | Dataset | SOGₖ | w/o SOGₖ |
> | --------- | ------- | ----------- |
> | BBBP    | 39.5  | 18.6      |
> | Tox21   | 56.2  | 23.4      |
> | ClinTox | 52.1  | 39.9      |
> | HIV     | 61.9  | 32.7      |
> | BACE    | 47.5  | 20.8      |
>
> We also provide specific results on all 17 tasks.
>
> | Task                   | SOGₖ | w/o SOGₖ |
> | ------------------------ | ------- | ----------- |
> | BBBP\_p\_np            | 39.5  | 18.6      |
> | Tox21\_NR-AR           | 63.9  | 26.9      |
> | Tox21\_NR-AR-LBD       | 57.8  | 19.9      |
> | Tox21\_NR-AhR          | 55.7  | 23.1      |
> | Tox21\_NR-Aromatase    | 60.0    | 26.7      |
> | Tox21\_NR-ER           | 59.8  | 34.2      |
> | Tox21\_NR-ER-LBD       | 51.9  | 23.2      |
> | Tox21\_NR-PPAR-gamma   | 55.5  | 19.1      |
> | Tox21\_SR-ARE          | 56.1  | 20.9      |
> | Tox21\_SR-ATAD5        | 58.1  | 23.5      |
> | Tox21\_SR-HSE          | 45.7  | 18.3      |
> | Tox21\_SR-MMP          | 53.8  | 22.3      |
> | Tox21\_SR-p53          | 55.7  | 22.3      |
> | ClinTox\_FDA\_APPROVED | 41.8  | 31.6      |
> | ClinTox\_CT\_TOX       | 62.3  | 48.3      |
> | HIV\_HIV\_active       | 61.9  | 32.7      |
> | BACE\_Class            | 47.5  | 20.8      |
>
> Also, small LLMs (eg.LLaMA3-3B) have limited out-of-the-box generalization (compared to large LLMs such as 175B GPT-3.5-turbo). Fine-tuning is​ to **ensure fair comparisons** with baselines as a standard evaluation protocol for small LLMs.

---

### Official Review · Reviewer_qiqx · 2025-11-01

**Soundness:** 3
**Presentation:** 3
**Contribution:** 3
**Rating:** 4
**Confidence:** 3

**Summary:**

The authors propose to use a special token <SOG_k> to represent the entire topology of a graph, so that LLM can use that token to make predictions. The method builds a structural tokenizer using GNNs and quantizes the representation into one entry. In addition, the authors construct hybrid structure QA corpora to align the new structural tokens with text tokens using LoRA. On five MoleculeNet-style graph-level classification datasets, their method improves over zero-shot/few-shot/LoRA/soft-prompt LLaMA baselines by a large margin.

**Strengths:**

1. The problem is clearly defined with proper related works.
2. Using one token for each graph is a convincing idea.
3. The overall tokenizer pipeline is clear and meaningful.
3. The authors show strong empirical gains on 3B and 7B models.

**Weaknesses:**

1. All main experiments are on small to moderate molecular graphs, where graph sizes and motif variety are relatively small. It’s unclear how this scales to large real-world graphs.
2. Compressing a large graph into a single token may lose information.
3. I did not see the training cost analysis.
4. Hyperparameter sweeping for the baseline is not reported.

**Questions:**

1. What is the exact size, structural diversity, or negative-sampling strategy of those three QA corpora?
2. Is there any out-of-vocabulary structure case?

---

> ### Author Response · Authors · 2025-11-25
>
> # Response to qiqx (1/5)
>
> We sincerely thank you for the constructive feedback and for recognizing the clarity of our problem definition, meaningful tokenizer design and alignment strategy, as well as the strength of our empirical results across 3B and 7B models. We especially appreciate the acknowledgment of our core idea of using one token for LLM structural understanding. Also, your concerns are important points, and we address each in the responses below. We hope our clarifications and additional analyses can better convey the motivation, robustness, and generality of our proposed method.
>
> #### ​**W1**​:
>
> Thank you for your comments. We fully agree with the point that generalizing to different larger-scale graphs is important. Besides molecular graphs, ​**we have already presented our evaluations on citation networks, i.e., Cora (2,485 nodes) and Pubmed (19,717 nodes) in Tab.3**​. To further evaluate our effectiveness, we also take your advice and supplement more experiments on citation network **OGBN-Arxiv (169,343 nodes)** and sentiment analysis dataset​ **SST-2 (tree-structured patterns)** ​, comparing with​ ​publicly reported performance in related literature[1,2,3]. The results demonstrate that our method also works effectively on substantially larger graphs and across different domains.
>
> | OGBN-Arxiv       |              |               |
> | ------------------ | -------------- | --------------- |
> | Technique        | Methods      | Accuracy      |
> | GNN-based        | GCN          | 72.24         |
> |                  | GAT          | 71.85         |
> |                  | GraphSAGE    | 71.88         |
> | LLM as enhancer  | GIANT        | 72.04         |
> |                  | TAPE         | 72.99         |
> |                  | OFA          | 73.23         |
> |                  | ENGINE       | 75.01         |
> | LLM as predictor | GraphAdapter | 74.45         |
> |                  | LLaGA        | 72.78         |
> |                  | InstructGLM  | 39.09         |
> |                  | GraphText    | 49.47         |
> |                  | SOGₖ        | __78.22__ |
>
> | SST-2     |                        |          |
> | ----------- | ------------------------ | ---------- |
> | Technique | Methods                | Accuracy |
> | GNN-based | ERM                    | 79.3     |
> |           | ASAP                   | 81.3     |
> |           | IRM                    | 81.1     |
> |           | Group DRO              | 81.1     |
> |           | GIL                    | 83.4     |
> |           | LiSA                   | 80.4     |
> | LLM-based | GPT2-base              | 50.2     |
> |           | GPT2-large             | 63.3     |
> |           | OPT-6.7B               | 93.5     |
> |           | OPT-13B                | 96.0       |
> |           | InstructGPT-3          | 91.6     |
> |           | InstructGPT-3-few-shot | 92.4     |
> |           | InstructGPT-3-CoT      | 92.1     |
> | LLaMA3-3B | LLaMA3-3B              | 65.7     |
> |           | LLaMA3-3B-few-shot     | 67.3     |
> |           | SOGₖ (LLaMA3-3B)      | __98.7__     |
>
> Due to time constraints, we performed downsampling during the LLM fine-tuning stage, training the model on only 2% of the original training data selected at random. For evaluation, we report results on a randomly sampled subset of 973 instances from the test set.
>
> [1] Li, Yuhan, et al. "Glbench: A comprehensive benchmark for graph with large language models." *Advances in Neural Information Processing Systems* 37 (2024): 42349-42368.
>
> [2] Tang, Ruixiang, Dehan Kong, and Longtao Huang. "Large language models can be lazy learners: Analyze shortcuts in in-context learning." ​*Findings of the Association for Computational Linguistics: ACL 2023*​. 2023.
>
> [3] Sun, Xiaofei, et al. "Text classification via large language models." *arXiv preprint arXiv:2305.08377* (2023).

---

> > ### Author Response · Authors · 2025-11-25
> >
> > # Response to qiqx (2/5)
> > #### ​**W2**​:
> >
> > We appreciate your insightful concern. Using **a single token** to represent an entire graph is indeed our ​*optimal goal*​, aiming for the most compact and LLM-friendly structural encoding. In this paper, we empirically show that such a design is **feasible and promising. ​**We observe that even with **only one ​**structural token,  <SOGₖ> achieves SOTA performance across multiple benchmarks.
> >
> > To be frank, we acknowledge that, as you've pointed out, compressing may lose certain information. Another reviewer (g287) raises similar concerns (​*has the authors explored using multiple structural tokens (e.g., 2–5 tokens) to capture richer or hierarchical graph properties?* ​), and we perform additional experiments. The results show that, in some cases, such as BBBP and BACE dataset containing more functional groups, using *more than one* token indeed yields improvements. However, in the majority of datasets, we find that the ​**single-token design performs best overall**​, offering a favorable balance between structural fidelity, token efficiency, and LLM alignment.
> >
> > | LLaMA3.2-3B |                        |             |             |             |             |
> > | ------------- | ------------------------ | ------------- | ------------- | ------------- | ------------- |
> > |             |                        | SOGₖ       | 2 tokens    | 3 tokens    | 5 tokens    |
> > | ClinTox     | ClinTox\_FDA\_APPROVED | 76.1 ± 5.3 | 74.1 ± 6.4 | 74.1 ± 3.2 | 64.8 ± 6.4 |
> > |             | ClinTox\_CT\_TOX       | 95.0 ± 0.1 | 98.3 ± 2.9 | 96.7 ± 5.8 | 96.7 ± 2.9 |
> > | HIV         | HIV\_HIV\_active       | 75.7 ± 1.6 | 56.5 ±0.4  | 56.3 ±1.2  | 56.4 ±0.2  |
> > | BACE        | BACE\_Class            | 63.3 ± 4.2 | 72.9 ±4.7  | 73.4 ±2.0  | 70.7 ±3.8  |
> > | BBBP        | BBBP\_p\_np            | 76.9 ± 3.1 | 81.5 ±1.4  | 81.3 ±1.1  | 81.2 ±1.1  |
> > |             | avg.                   | __77.4__        | 76.7        | 76.4        | 74.0          |
> >
> > In practical real-world deployments, standard engineering strategies such as ​**graph partitioning**​, ​**community detection**​, or **structural decomposition ​**can be used, allowing a large graph to be represented by ​**multiple structural tokens**​. This naturally extends our approach from 1 token to *K*  tokens when needed.

---

> > > ### Author Response · Authors · 2025-11-25
> > >
> > > # Response to qiqx (3/5)
> > > #### ​**W3**​:
> > >
> > > We sincerely thank you for the reminder. We have added a training cost analysis, which we hope further demonstrates that the computational cost of our method is both manageable and acceptable. '-' represents the same as above.
> > >
> > > |                  |         | #parameters | speed     | memory    | resource |
> > > | ------------------ | --------- | ------------- | ----------- | ----------- | ---------- |
> > > | SOGₖ(LLaMA3-3B) | stage 1 | 394,002,432 | 5.45it/s  | 35,282 MB | RTX4090  |
> > > |                  | stage 2 | 418,316,288 | 2.42it/s  | 17,118MB  | RTX4090  |
> > > | SOGₖ(LLaMA2-7B) | stage 1 | 264,241,152 | 10.83s/it | 66,390MB  | RTX3090  |
> > > |                  | stage 2 | 304,218,112 | 19.67s/it | 16,295MiB | RTX4090  |
> > >
> > > | Backbone  | Dataset      | Stage1 Pre-training time | Stage2 Pre-training time | Downstream Training time |
> > > | ----------- | -------------- | -------------------------- | -------------------------- | -------------------------- |
> > > | LLaMA3-3B | BBBP (graph) | 1.10h                    | 3.21h                    | 0.27h                    |
> > > |           | HIV (graph)  | -                        | -                        | 6.65h                    |
> > > |           | cora (node)  | -                        | -                        | 0.48h                    |
> > > | LLaMA2-7B | BBBP (graph) | 3.57h                    | 6.95h                    | 0.47h                    |
> > > |           | HIV (graph)  | -                        | -                        | 10.35h                   |
> > > |           | cora (node)  | -                        | -                        | 2.08h                    |

---

> > > > ### Author Response · Authors · 2025-11-25
> > > >
> > > > # Response to qiqx (4/5)
> > > > #### ​**W4**​:
> > > >
> > > > We appreciate your valuable suggestion. Following your advice, we have now included a hyperparameter analysis, as presented below, evaluating the structural vocabulary size K in different values (e.g., K=64, 128, 256, 512).
> > > >
> > > > From the average results, we observe that **K = 256** yields the strongest overall performance. When *K*  is too small, many structural patterns cannot be sufficiently distinguished; when *K*  is too large, the model tends to overfit and loses generalization ability. **K = 256 ​** provides an effective balance.
> > > >
> > > > | Dataset | K = 64 | K = 128 | K = 256\* | K = 512 |
> > > > | --------- | -------- | --------- | ----------- | --------- |
> > > > | BBBP    | 84.6   | 71.8    | 76.9      | 96.2    |
> > > > | ClinTox | 91.8   | 90.1    | 76.9      | 90.0      |
> > > > | HIV     | 52.7   | 51.9    | 79.7      | 55.9    |
> > > > | BACE    | 69.8   | 62.3    | 98.4      | 85.7    |
> > > > | Tox21   | 52.6   | 53.4    | 81.6      | 59.8    |
> > > > | __avg.__    | __60.1__   | __59.3__    | __81.7__      | __66.8__    |
> > > >
> > > > We also provide specific AUC-ROC results for each 17 tasks.
> > > >
> > > > | Task                   | K = 64      | K = 128     | K = 256\*   | K = 512      |
> > > > | ------------------------ | ------------- | ------------- | ------------- | -------------- |
> > > > | BBBP\_p\_np            | 84.6 ± 3.2 | 71.8 ± 3.4 | 76.9 ± 3.1 | 96.2 ± 0.8  |
> > > > | ClinTox\_FDA\_APPROVED | 95.2 ± 2.9 | 92.5 ± 0.2 | 77.8 ± 4.1 | 100.0 ± 0.0 |
> > > > | ClinTox\_CT\_TOX       | 88.5 ± 2.5 | 87.6 ± 3.8 | 76.1 ± 3.4 | 80.0 ± 10.0 |
> > > > | HIV\_HIV\_active       | 52.7 ± 0.0 | 51.9 ± 0.4 | 79.7 ± 3.0 | 55.9 ± 2.0  |
> > > > | BACE\_Class            | 69.8 ± 6.1 | 62.3 ± 2.1 | 98.4 ± 0.9 | 85.7 ± 1.7  |
> > > > | Tox21\_NR-AR           | 60.7 ± 3.7 | 61.3 ± 2.2 | 76.4 ± 2.1 | 60.9 ± 1.8  |
> > > > | Tox21\_NR-AR-LBD       | 58.5 ± 1.5 | 58.6 ± 6.5 | 76.0 ± 6.7 | 55.2 ± 2.6  |
> > > > | Tox21\_NR-AhR          | 55.4 ± 0.3 | 56.2 ± 1.0 | 94.1 ± 2.9 | 61.8 ± 1.1  |
> > > > | Tox21\_NR-Aromatase    | 50.0 ± 0.0 | 50.3 ± 0.6 | 82.3 ± 1.4 | 70.6 ± 2.2  |
> > > > | Tox21\_NR-ER           | 51.6 ± 0.3 | 55.8 ± 1.6 | 84.9 ± 4.1 | 67.6 ± 2.6  |
> > > > | Tox21\_NR-ER-LBD       | 52.2 ± 0.1 | 53.8 ± 1.4 | 87.8 ± 2.6 | 60.9 ± 2.7  |
> > > > | Tox21\_NR-PPAR-gamma   | 50.0 ± 0.0 | 51.5 ± 1.3 | 82.8 ± 1.5 | 56.8 ± 2.3  |
> > > > | Tox21\_SR-ARE          | 50.3 ± 1.3 | 50.6 ± 1.8 | 85.2 ± 2.6 | 57.5 ± 2.0  |
> > > > | Tox21\_SR-ATAD5        | 49.9 ± 0.1 | 50.0 ± 0.0 | 76.1 ± 5.3 | 59.0 ± 3.1  |
> > > > | Tox21\_SR-HSE          | 50.7 ± 0.6 | 50.0 ± 0.0 | 95.0 ± 0.1 | 52.1 ± 1.0  |
> > > > | Tox21\_SR-MMP          | 51.7 ± 1.0 | 52.8 ± 2.2 | 75.7 ± 1.6 | 57.3 ± 0.4  |
> > > > | Tox21\_SR-p53          | 50.0 ± 0.0 | 50.6 ± 0.6 | 63.3 ± 4.2 | 57.6 ± 1.8  |
> > > > | __avg.__                  | __60.1__        | __59.3__        | __81.7__        | __66.8__         |

---

> > > > > ### Author Response · Authors · 2025-11-25
> > > > >
> > > > > # Response to qiqx (5/5)
> > > > > #### ​**Q1**​:**​ ​**
> > > > >
> > > > > Thank you for your helpful suggestion, and we apologize for not making this point sufficiently clear in the original manuscript. We clarify the size, structural diversity, and negative-sampling strategy of the three QA corpora below.
> > > > >
> > > > > ​**k-Nearest Token Neighbour Matching QA**​: For each of the 256 tokens, we identify the five nearest and five farthest tokens in the graph learning space. Using two different prompt templates, a total of 1024 data points are generated.
> > > > >
> > > > > ​**True/False Structure Similarity Judgment QA**​: We calculate the cosine similarity of the 256 tokens in the graph learning space. Tokens with similarity greater than 0.8 are considered positive samples, while less than 0.2 are negative. A total of 14,899 samples are constructed, consisting of 8,452 positives and 6,447 negatives.
> > > > >
> > > > > ​**Description-Token Pairs Matching QA**​: We include a total of 53,975 graphs from MoleculeNet and 2,485 ego-graphs from citation network. Each data point consists of a natural language description of the pure structure and corresponding structural token.
> > > > >
> > > > > #### ​**Q2**​:**​ ​**
> > > > >
> > > > > Thank you for this important question. While such cases do exist, they do not affect overall performance. For example, in the test set, among 5,290 molecule skeletons, only **33** skeletons (​ **∼0.6%** ​) exhibit inconsistency (molecules with same skeleton are mapped to different tokens). However, this issue is further mitigated by the Description-Token Pairs Matching QA alignment (Sec 4.2), which encourages LLM to internalize diverse structural patterns within a single token embedding for out of vocabulary toleration.

---

### Official Review · Reviewer_kfpk · 2025-11-03

**Soundness:** 2
**Presentation:** 3
**Contribution:** 2
**Rating:** 4
**Confidence:** 4

**Summary:**

As LLMs are facing challenges with graphs because of their structural hallucination. Existing approaches have their apparent shortages. This paper proposes a new method that insert a special token called  <SOGk > to represent the structure of graph better. In this paper, the author proposes a special tokenizer to generate a highly selective token based on graph topology, then align the new <SOGk > token with the existing tokens by a set of hybrid structure Question-Answering corpora. The experiments results show it significantly helps LLMs to understand graph datasets better.

**Strengths:**

This paper innovatively creates a new method in the graph recognition of large language models (LLMs). Unlike previous methods, which usually convert the entire graph data into another format, the Structure - Oriented Graph (SOG) provides a novel way for LLMs to understand textual graphs through a special token that reflects the overall structure of the graph.

The advantage of SOG is that it uses a Graph Neural Network (GNN) as an encoder to obtain deeper information within the graph data. To address the drawbacks in converting continuous embeddings, SOG maps the continuous representations into a K - discrete token vocabulary, which is helpful in solving misalignment. It seems to be quite original.

**Weaknesses:**

Method proof missing: This paper only elaborates the main idea of SOG but lacks some basic proof regarding how this method emerged. From an objective perspective, some basic algorithms need to be listed to prove how the components of SOG evidently improve the accuracy and efficiency in graph learning.

The Hybrid QA method is not always the best way to align a new single token with existing tokens. The chart in this paper shows that in some cases (BBBP, ClinTox), Hybrid QAs are slightly worse than KNN matching alone. This indicates that SOG still needs to find a better way to align its structural token with other tokens on more complex datasets, especially when fine - tuning with description - token.

**Questions:**

NA

---

> ### Author Response · Authors · 2025-11-25
>
> # Response to kfpk (1/2)
>
> We sincerely thank you for the thoughtful and detailed feedback. We are grateful for the recognition of our method’s originality in representing graph structure through a dedicated structural token, as well as the potential of our tokenizer and alignment strategy to improve LLMs' understanding of graphs. Meanwhile, we address each of your concerns in the responses below. We hope the clarifications and additional results will help better illustrate the motivation, design, and effectiveness of our method.
>
> #### **Q1: ​**
>
> Thank you for your comment.**​ ​**Our key innovation is to incorporate only one special token <SOGₖ> for LLM to fully represent the graph structure. It is instantiated ​**through two key components**​: **Topology-Aware Graph Structural Tokenizer** and ​**Hybrid Structure QAs Token Alignment**​. The **Structural Tokenizer** extracts and maps each graph structure into a highly selective single token​ **​ <SOGₖ>** ​, and the **Hybrid QAs Alignment ​**instruct LLM to understand, generate and reason with <SOGₖ> like textual tokens for a unified topology understanding. ​The design of these components is grounded in clear algorithmic motivations​—each step is introduced to progressively retain topology, compress structural information, and ensure compatibility with LLM tokenization.
>
> To demonstrate how the components **<SOGₖ>** and **Hybrid QAs ​**evidently improve accuracy and efficiency, we've provided a detailed ablation study in the original manuscript (in **Tab.2** and ​**Fig.3**​). Since they primarily focus on the interaction with LLMs, we supplement experiments on the proposed ​**Topology-Aware Graph Tokenizer**​. To be specific, we ablate three key components—​ **(i) anchor node selection**​, ​ **(ii) virtual global node**​, and **(iii) self-supervised topological reconstruction. ​**
>
> **(i) anchor node selection**
>
> In the original manuscript, we use node **degree** to select anchor node and locate other nodes accordingly. We further compare AUC-ROC on benchmarks with more anchor-selection strategies to examine its influence on structural encoding and downstream performance.
>
> |                        | random | pagerank | betweeness | degree\* |
> | ------------------------ | -------- | ---------- | ------------ | ---------- |
> | BBBP\_p\_np            | 74.4   | __82.2__     | 80.9       | 76.9     |
> | HIV\_HIV\_active       | 72.0     | 75.4     | 62.3       | __79.7__     |
> | BACE\_Class            | 63.5   | 64.3     | 65.5       | __98.4__     |
> | ClinTox\_FDA\_APPROVED | 92.4   | __99.6__     | 98.0         | 77.8     |
> | Tox21\_NR-AR           | 69.6   | 59.6     | 69.7       | __76.4__     |
> | Tox21\_NR-AR-LBD       | 66.6   | 58.7     | 64.0        | __76.0__       |
> | Tox21\_SR-p53          | __66.6__   | 54.6     | 56.4       | 63.3     |
>
> We find that in most cases, the simple and effective **degree** strategy remains the best choice. For BBBP and ClinTox, **PageRank** performs slightly better because it better captures global centrality under sparse molecular graphs. Conversely, in most settings, **random** anchor selection shifts the structural center of graphs unpredictably, leading to unstable structural encoding and degraded performance.
>
> **(ii) virtual global node**
>
> Our original design uses a **global node** to perform structural pooling. We eliminate the global node (use anchor node instead) to assess influences of structural information aggregation strategy.
>
> | Dataset | w/o global node | original |
> | --------- | ----------------- | ---------- |
> | HIV     | 52.8            | __79.7__     |
> | BACE    | 54.9            | __98.4__     |
> | Tox21   | 53.7            | __81.6__     |
>
> We also provide AUC-ROC results for each task specifically.
>
> | Task                 | w/o global node | original |
> | ---------------------- | ----------------- | ---------- |
> | HIV\_HIV\_active     | 52.8            | 79.7     |
> | BACE\_Class          | 54.9            | 98.4     |
> | Tox21\_NR-AR         | 61.8            | 76.4     |
> | Tox21\_NR-AR-LBD     | 57.6            | 76.0       |
> | Tox21\_NR-AhR        | 58.1            | 94.1     |
> | Tox21\_NR-Aromatase  | 50.5            | 82.3     |
> | Tox21\_NR-ER         | 57.7            | 84.9     |
> | Tox21\_NR-ER-LBD     | 51.9            | 87.8     |
> | Tox21\_NR-PPAR-gamma | 52.3            | 82.8     |
> | Tox21\_SR-ARE        | 51.6            | 85.2     |
> | Tox21\_SR-ATAD5      | 51.0              | 76.1     |
> | Tox21\_SR-HSE        | 49.8            | 95.0       |
> | Tox21\_SR-MMP        | 51.7            | 75.7     |
> | Tox21\_SR-p53        | 50.6            | 63.3     |
>
> We observe a significant degradation in AUC-ROC performance when the global node is removed, demonstrating its effectiveness in capturing and compressing essential structural information ​in a concise and efficient manner​.

---

> ### Author Response · Authors · 2025-11-25
>
> # Response to kfpk (2/2)
>
> **(iii) self-supervised topological reconstruction**
>
> The tokenizer’s training loss contains a self-supervised reconstruction term. To assess its impact, we remove this loss (keeping only the warmup step) and evaluate the resulting performance.
>
> | Dataset | w/o reconstruction | original |
> | --------- | -------------------- | ---------- |
> | HIV     | 71.5               | __79.7__     |
> | BACE    | 66.1               | __98.4__     |
> | BBBP    | __78.9__               | 76.9     |
> | Tox21   | 54.0                 | __81.6__     |
>
> We also provide AUC-ROC results for each task specifically.
>
> | Task                 | w/o reconstruction | original |
> | ---------------------- | -------------------- | ---------- |
> | HIV\_HIV\_active     | 71.5               | 79.7     |
> | BACE\_Class          | 66.1               | 98.4     |
> | BBBP\_p\_np          | 78.9               | 76.9     |
> | Tox21\_NR-AR         | 59.7               | 76.4     |
> | Tox21\_NR-AR-LBD     | 62.1               | 76.0       |
> | Tox21\_NR-AhR        | 60.4               | 94.1     |
> | Tox21\_NR-Aromatase  | 50.0                 | 82.3     |
> | Tox21\_NR-ER         | 54.5               | 84.9     |
> | Tox21\_NR-ER-LBD     | 53.8               | 87.8     |
> | Tox21\_NR-PPAR-gamma | 52.3               | 82.8     |
> | Tox21\_SR-ARE        | 50.6               | 85.2     |
> | Tox21\_SR-ATAD5      | 50.5               | 76.1     |
> | Tox21\_SR-HSE        | 51.1               | 95.0       |
> | Tox21\_SR-MMP        | 52.8               | 75.7     |
> | Tox21\_SR-p53        | 50.5               | 63.3     |
>
> After removing the reconstruction loss, the performance generally degrades, with a dramatic **27.6** AUC-ROC drop on Tox21. The absence of the self-supervised reconstruction component prevents the tokenizer from adequately capturing structural information and embedding it into the structural vocabulary.
>
> #### **Q2: ​**
>
> Thank you for your comment. As you have correctly pointed out, our experimental results indeed show some suboptimal cases. However, taking into account all different datasets, ​**Hybrid QAs demonstrate clear overall superiority**​. It is worth noting that Hybrid QAs consistently yield better overall results, i.e., **+6.2/+7.9** (LLaMA3-3B) and **+2.9/+2.8** (LLaMA2-7B) AUC-ROC improvements over KNN-only and KNN+T/F Judgement on average. This suggests that Hybrid QA is especially beneficial for real-world applications where structural complexity is higher or unknown, as it combines complementary alignment signals to enhance structural understanding. In practice, KNN can be effective in simple cases occasionally, but Hybrid QA offers greater robustness and generalization.

---

### Author Response · Authors · 2025-11-25

# General Response by Authors

We thank all the reviewers for their thoughtful and constructive feedback, as well as their recognition of the potential and contributions of our work. We are greatly encouraged by their positive assessments, which we summarize as follows:

1. **Innovation and Conceptual Elegance**

Multiple reviewers highlighted the novelty and elegance of our core idea -- utilizing only one structural token to enhance the LLM topology understanding.

* Specifically, **Reviewer kfpk** stated that our method *​“innovatively creates a new method in the graph recognition of large language models (LLMs)”​* and provides a ​ *“novel way for LLMs to understand textual graphs”* ​.
* **Reviewer g287** further elaborated on this, describing it as an *“innovative and elegant design: Representing an entire graph via a single virtual global node mapped to a vocabulary index is conceptually clean and computationally efficient.”*

2. **Effectiveness and Strong Empirical Performance**

All reviewers acknowledged the practical advantages and performance gains of our proposed <SOGₖ> approach.

* **Reviewer tD9C** noted that it achieves *“large improvements on multiple MoleculeNet tasks with clear ablations,”*
* **Reviewer qiqx** found the results convincing, stating that *“the authors show strong empirical gains on 3B and 7B models.”*
* **Reviewer g287** also emphasized the method's *​“high efficiency and strong performance,”​* achieving competitive results with *“minimal token overhead (just +1 token) and negligible parameter cost.” ​*
* The practical benefit of token efficiency was underscored by ​**Reviewer tD9C**​, who highlighted that it *​“drastically reducing prompt length vs graph-to-text.”​* and *“works with off-the-shelf LLMs.”*

3. **Clarity and Interpretability of Analysis**

The clarity of the presentation and the inclusion of insightful analysis were also positively recognized.

* **Reviewer qiqx** found the problem definition and pipeline to be *​“clearly defined”​*  and *“clear and meaningful.”*
* Furthermore, **Reviewer tD9C** and **Reviewer g287** both valued the *​“interpretability analysis,”​* with the latter noting that it shows *“each <SOGₖ> token occupies a distinct region in the embedding space, suggesting meaningful structural encoding.”*

---

### Author Response · Authors · 2025-11-30

Dear PC, SAC, AC,

Thank you for reviewing this manuscript. As the deadline for the authors' response approaches, we have summarized this manuscript and rebuttal as follows.

# Summary of the manuscript

We propose to utilize **only one structural token <SOGₖ>** to effectively and selectively provide structural information to LLM, facilitating explicit topology input and structural information sharing. The overall framework consists of two stages: we design a **topology-aware graph structural tokenizer** to extract topology for one structural token and construct multiple **hybrid structure Question-Answer (QA) pairs ​**to harmonize text and topology for a unified understanding in LLM. With this approach, <SOGₖ> empowers LLMs to **understand, generate, and reason** topology in a concise and accurate manner.

# Reviwer Acknowledgement

We are truly encouraged to receive the positive feedback from the reviewers as we summarized **in the previous official comment**.
1. They highlighted the novelty and elegance of our core idea -- representing an entire graph with a single structural token to enhance LLM topology understanding​ ​(**Reviewer kfpk**, **Reviewer g287**)​.
2. They also acknowledged the practical advantages and strong empirical performance of our <SOGₖ> approach ​(**Reviewer tD9C**, **Reviewer qiqx**, **Reviewer g287**, **Reviewer tD9C**)​.
3. The clarity and insightfulness of our analysis were also positively recognized.​(**Reviewer qiqx**, **Reviewer tD9C**, **Reviewer g287**)​.

# Summary of the rebuttal

We appreciate the constructive engagement from all reviewers throughout this process. The overall rating of this manuscript is 6, 4, 4, 4. Although we did not receive follow-up replies during the rebuttal phase, we remain sincerely grateful for the reviewers’ thoughtful comments and suggestions, which have greatly helped us refine and improve the manuscript.

We have actively addressed the reviewers’ comments by conducting a series of new analyses and experiments, ***the revised portions have been highlighted in red in the latest submitted manuscript***:

​**First**​, we supplement ablations on three key components and four anchor node choices focusing our Topology-Aware Graph Tokenizer (**Reviewer kfpk, Reviewer g287**)​.

​**Second**​, we extend to substantially larger and structurally different datasets and achieve SOTA accuracy to demonstrate strong effectiveness across domains ​(**Reviewer qiqx, Reviewer tD9C**)​.

​**Third**​, we test multi-token variants which further confirm that the single-token design offers the best overall trade-off while remaining easily extensible when needed (​**Reviewer qiqx, Reviewer g287**)​.

​**Fourth**​, we incorporate a training cost analysis to demonstrate that the computational overhead of our approach is manageable ​(**Reviewer qiqx**​). ​

**Fifth**​, we include a sensitivity analysis of the structural vocabulary size K, showing that our choice of K = 256 provides the strongest balance between expressiveness and generalization ​(**Reviewer qiqx, Reviewer g287**​).​

**Finally**​, we add several clarifying details, such as dataset scales ​(**Reviewer qiqx**​), OOV cases (**Reviewer qiqx**) , anchor bias (**Reviewer tD9C**) and include comparisons with very recent methods (**Reviewer tD9C, Reviewer g287**) to ensure completeness and transparency.

We believe that through revisions to the manuscript and thorough communication with the reviewers, **we have effectively addressed their concerns, and the manuscript is now presented in a more complete and convincing manner**.

The above is our summary of this article and rebuttal. Thank you again to PC, SAC, and AC for the time and effort in reviewing this manuscript.

---

### Meta-Review · Area_Chair_UkbG · 2026-01-07

**Summary:**

This paper designs a tokenizer for graph structure learning.

**Reviewer Concerns:**

Most of the reviewer concerns were on experiments on terms of datasets, baseline, analysis. The authors provided extensive results during rebuttals and it seems these concerns have been addressed.

**Reviewer Scores:**

Reviewers scores are on the lower side, but I feel they would raise if allowed.

---

### Decision · Program_Chairs · 2026-01-26

Accept (Poster)